# Convergent evolution of distinct D-ribulose utilisation pathways in attaching and effacing pathogens

Curtis Cottam [1], Kieran Bowran [1], Rhys T. White [2], Arnaud Baslé[1], Inokentijs Josts[1] & James P. R. Connolly [1] ✉

Attaching and effacing pathogens overcome colonisation resistance by competing with metabolically similar organisms for limited resources. Enterohaemorrhagic *E. coli* (EHEC) utilises the pathogenicity island-encoded Accessory L-arabinose Uptake (Aau) transporter to effectively colonise the mouse gut, hypothesised to be achieved via an enhanced capacity to scavenge L-arabinose. Aau is regulated exclusively in response to L-arabinose, but it is unclear how this system specifically benefits EHEC in vivo. Here, we show that Aau displays a > 200-fold higher affinity for the monosaccharide D-ribulose, over L-arabinose. EHEC cannot grow on D-ribulose as a sole carbon source and this sugar does not trigger *aau* transcription. However, Aau effectively transports D-ribulose into the cell only in the presence of L-arabinose, where it feeds into the pentose phosphate pathway, after phosphorylation by the L-ribulokinase AraB, thus providing EHEC a significant fitness advantage. EHEC has therefore evolved a mechanism of hijacking the canonical L-arabinose utilisation machinery to promote D-ribulose utilisation in vivo. Furthermore, *Citrobacter rodentium* encodes an analogous system that exclusively transports D-ribulose and metabolises it via a dedicated D-ribulokinase. These unique mechanisms of D-ribulose utilisation suggest that convergent evolution has driven the ability of distinct pathogenic species to exploit this nutrient during invasion of the gut niche.

Enteric pathogens utilise numerous strategies to overcome colonisation resistance imposed by the native gut microbiota[1]. For example, diverse virulence factors are often encoded on horizontally acquired genomic elements and are critical for infection[2]. Traditional virulence factors usually facilitate physical interactions in the host environment, for instance, adhesion to the host cell surface, immune modulation by effector proteins, or interbacterial killing by toxins[3,4]. However, bacterial cells fundamentally need energy for growth, regardless of whether they are pathogenic or not[5]. Therefore, employing unique strategies to scavenge limited nutrients in the gut and outcompete native species with similar metabolic requirements is a critical trait of pathogens.

ATP-binding cassette (ABC) transporters are ubiquitous across all domains of life and represent the largest and most ancient protein superfamily. They facilitate the transport of a diverse range of substrates across biological membranes, including ions, sugars, lipids and peptides[6]. ABC transporters consist of two cytoplasmic nucleotide-binding domains and two variable transmembrane domains forming a modular architecture[7]. In addition, substrate binding proteins (SBPs) are typically encoded at the same locus and capture target substrates, dictating the specificity of the cognate transporter[8]. Although primarily associated with the acquisition of nutrients for normal cellular function and growth, ABC transporters have also been

[1]Newcastle University Biosciences Institute, Newcastle University, Newcastle-upon-Tyne, UK. [2]New Zealand Institute for Public Health and Forensic Science, Health Security, Porirua, New Zealand. ✉e-mail: James.Connolly2@newcastle.ac.uk

implicated in the pathogenesis of clinically significant species such as *Escherichia coli* [9]. Distinct pathotypes of *E. coli* frequently encode numerous accessory ABC transport systems on genomic islands not found in commensal strains [10,11]. These systems are often upregulated in vivo and show specificity for key nutrients such as simple sugars, amino acids or co-factors, thus enhancing the ability to scavenge preferred sources of energy and exploit highly contested nutrient niches [12–16]. In addition, transported substrates can also modulate virulence gene expression downstream of their uptake into the cell. For example, we recently identified the accessory L-arabinose uptake (Aau) system in the attaching and effacing pathotype enterohaemorrhagic *E. coli* (EHEC). Aau is encoded widely by EHEC strains on a specific genomic island and is activated exclusively in response to L-arabinose-mediated regulation, enabling enhanced growth on this sugar and a competitive advantage in the murine gut [17]. Furthermore, downstream metabolism of L-arabinose was found to enhance expression of the LEE pathogenicity island-encoded type 3 secretion system, which is essential for EHEC colonisation of host cells [17–19]. Therefore, ABC transporters have emerged as important virulence factors, and elucidating their associated substrates and function could provide valuable insights into the competitive strategies employed by *E. coli* pathotypes in the host gut.

Here, we identify that uptake and metabolism of D-ribulose via specific ABC transport systems dramatically enhances the fitness of EHEC and the murine pathogen *Citrobacter rodentium*, widely adopted as the surrogate model for studying EHEC pathogenesis in vivo [20–22]. We show that convergent evolutionary pathways towards D-ribulose utilisation have emerged in both species. *C. rodentium* utilises a dedicated D-ribulose uptake and metabolism locus to process this sugar, whereas EHEC exploits the Aau transporter and functional flexibility in the canonical L-arabinose utilisation machinery to metabolise D-ribulose. These mechanistic insights identify D-ribulose as a potentially key nutrient that dictates the ability of pathogens to overcome colonisation resistance.

## Results

### *C. rodentium* encodes a highly specific D-ribulose utilisation locus (Rbl)

We recently reported the identification of an ABC transporter locus (encoded by *ROD_24811-41*) in *C. rodentium* that includes some of the most highly upregulated genes during murine infection (>15-fold) when compared to growth in laboratory media (Fig. 1a)[16,17]. This discovery led us to identify the analogous Aau system in EHEC, which shared similar predicted functionality but displayed a unique genomic context, suggesting they are independent systems of distinct origin (Fig. 1b, c). While sharing ~60 % identity in amino acid sequence across the substrate binding, ATPase and permease subunits of the transporters, one striking difference was that ROD_24811-41 was encoded in an operon alongside a predicted kinase (ROD_24851) and isomerase (ROD_24861) (Fig. 1b, c). This distinction and the difference in genomic context left the substrate specificity for ROD_24811-41 ambiguous and prompted us to investigate the function of this system in *C. rodentium*.

Domain analysis of ROD_24811 using InterProScan returned hits specific for SBP family II (SPB_2 domain; IPR025997) known to encompass proteins with specificity for simple monosaccharide substrates, including, but not exclusively, L-arabinose, D-ribose, D-xylose, and D-galactose. ROD_24851 is annotated as a D-ribulokinase, and InterProScan analysis confirmed the presence of two predicted actin-like ATPase domains (FGGY_N; IPR018484 and FGGY_C; IPR018485) characteristically found in members of the FGGY carbohydrate kinase family. In addition, ROD_24861 possessed a KpsF-like sugar isomerase domain (IPR035474), which is found in the D-arabinose-5-phosphate isomerase specific for D-ribulose-5-phosphate in *E. coli*. We therefore hypothesised that the ROD_24811-61 locus may mediate the transport and metabolic processing of D-ribulose as its substrate.

In support of this, transcriptional analysis using a *ROD_24811-61* reporter plasmid (pMK1*lux*-P$_{24811}$) revealed that D-ribulose significantly (up to 15-fold; $P = 0.0074$) activated expression of this system in a concentration dependent manner (Fig. 1d and Supplementary Fig. 1a) and we found that *C. rodentium* could grow in M9 minimal media supplemented with D-ribulose at concentrations as low as 0.05 mg/ml (Supplementary Fig. 1b). Activation of this system was found to be specific to D-ribulose, with L-ribulose failing to elicit any transcriptional response (Supplementary Fig. 1c). Furthermore, growth in the presence of L-arabinose and D-xylose failed to trigger any detectable pMK1*lux*-P$_{24811}$ activity, while D-ribose elicited a weak response that was 3-fold lower than that of D-ribulose when comparing the maximal luminescence recorded for each sugar (Fig. 1d). Crucially, growth analysis of wild type *C. rodentium* compared to a *ROD_24811-61* deletion mutant revealed that loss of this locus completely abolished its ability to grow on D-ribulose as a sole carbon source, a phenotype that was rescued by complementation with a plasmid constitutively expressing ROD_24811-61 (Fig. 1e). It was also noted that no growth defect was observed when this mutant was cultured on D-ribose as a sole carbon source (Supplementary Fig. 2). These data led us to conclude that *ROD_24811-61* encodes a utilisation system highly specific for D-ribulose, which we named Rbl herein for simplicity.

To determine the role of the individual Rbl components in D-ribulose utilisation, we generated deletions in genes encoding the ABC transporter (*ROD_24811-41*; termed *rblABCD*), D-ribulokinase (*ROD_24851*; termed *rblK*), and isomerase (*ROD_24861*; termed *rblI*) components. As with the entire locus deletion, no growth was observed for loss of either the transporter or D-ribulokinase, whereas loss of the isomerase had no growth defect (Supplementary Fig. 3). Lastly, our recent study on Aau found that uptake and metabolism of pentose sugars by EHEC results in the upregulation of the LEE-encoded type 3 secretion system via the generation of excess cellular pyruvate as a signal for gene regulation [17,23]. Therefore, in addition to growth analysis, we analysed whether D-ribulose could regulate virulence gene expression in *C. rodentium*. Using a *C. rodentium*-specific type three secretion system reporter (pMK1*lux*-P$_{LEE1Cr}$), we found that metabolism of D-ribulose results in significantly enhanced LEE transcription ($P = 0.0385$ at peak expression; Supplementary Fig. 4), suggesting that metabolism of this sugar would similarly promote virulence factor expression. These data collectively indicate that Rbl encodes a transport system and kinase that are essential for the utilisation of D-ribulose by *C. rodentium*, which enhances the pathogens fitness and virulence potential.

### The Aau transporter from EHEC displays high affinity for D-ribulose

Our functional data on D-ribulose utilisation by *C. rodentium* suggested that despite similarities between Rbl and the Aau transporter of EHEC, their roles in substrate recognition are distinct. However, while our previous in vivo studies on Aau identified an important role for the transporter during colonisation of the murine gut, the details of its exact substrate were somewhat ambiguous given that in vitro it appeared to have only a weak growth phenotype related to L-arabinose (that is, a partially elevated growth rate when Aau was overexpressed in a mutant for the canonical L-arabinose transporter AraE) [17]. We therefore reasoned that the SBP of Aau may have a broader substrate range. To test this, we overexpressed and purified a recombinant His-tagged version of the SBP from the EHEC Aau system (AauA; locus tag 0432 from strain ZAP193) by metal affinity and size exclusion chromatography (Supplementary Fig. 5). We then subjected purified AauA to Differential Scanning Fluorimetry (nanoDSF) thermal denaturation, using various monosaccharides as potential binding partners. This technique measures any increase in protein thermal stability, which would indicate a stabilising effect of the binding substrate. As expected, addition of D-glucose resulted in no thermal shift,

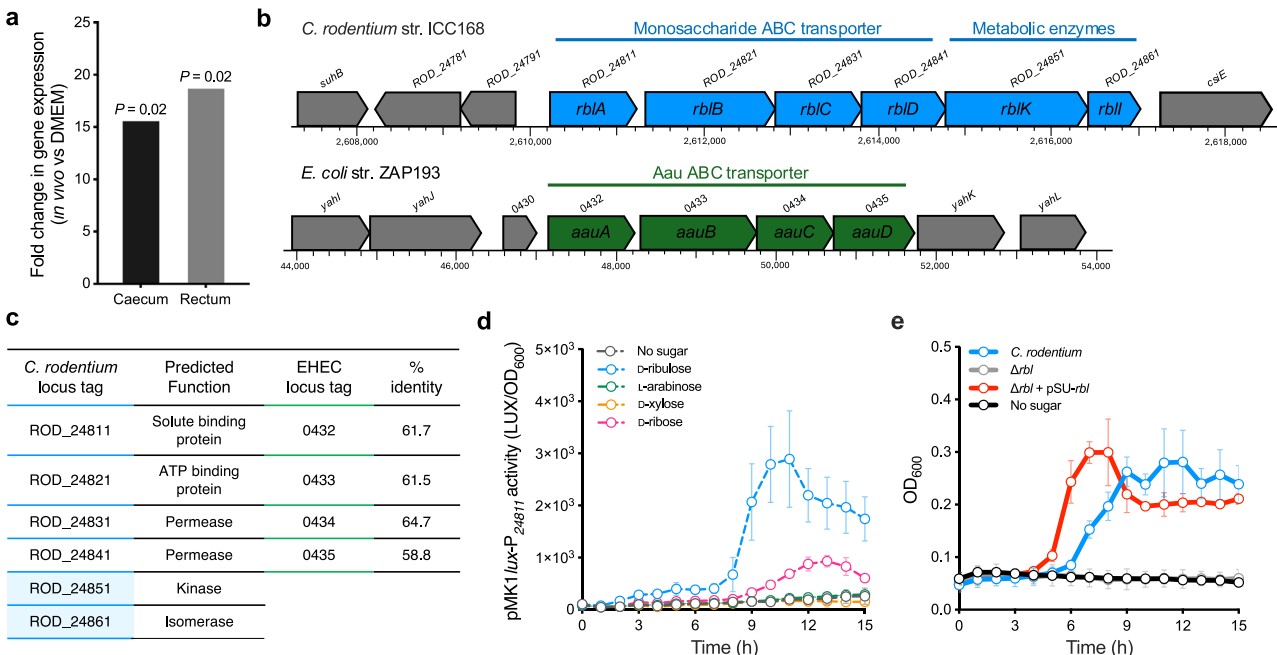

**Fig. 1 | *C. rodentium* encodes a highly specific D-ribulose utilisation system.**
**a** RNA-seq data identifying upregulation of *ROD_24811* during *C. rodentium* colonisation of the murine caecum and rectum at peak infection (day 10) compared to growth in laboratory medium (DMEM). Data were derived from Connolly et al.[16]. and represent differential expression calculated as fold change using edgeR with a false discovery rate of 5%. **b** Genomic context of the *C. rodentium ROD_24811-61* locus (blue) compared to the EHEC *aau* locus (green). Locus tags are labelled above each gene, with the proposed gene name illustrated within the corresponding gene. Chromosomal co-ordinates are indicated below and neighbouring genes in grey. **c** Summary of the predicted function and amino acid sequence percentage identity between the ROD_24811-61 and Aau components. **d** Transcriptional reporter assay

of *C. rodentium* transformed with the pMK1*lux*-P$_{24811}$ plasmid cultured in MEM-HEPES alone or supplemented with 0.5 mg/ml D-ribulose (blue), L-arabinose (green), D-ribose (pink) or D-xylose (orange). Data are depicted as luminescence units (LUX) divided by optical density (OD$_{600}$) of the culture at each timepoint. **e** Growth curve depicting OD$_{600}$ values over time of wild type *C. rodentium*, Δ*rbl* (full *ROD_24811-61* locus deletion), and Δ*rbl* + pSU-*rbl* cultured in M9 minimal media supplemented with 0.5 mg/mL D-ribulose. The no sugar control indicates wild type *C. rodentium* inoculated into M9 without a carbon source. Error bars for reporter assays and growth curves represent the standard deviation from the mean of three independent experiments (*n* = 3 biological replicates). Source data are provided as a Source Data file.

whereas assaying L-arabinose or D-ribose with AauA led to minor increases in melting temperature ($T_m$) of 2 °C over the buffer-only control (Fig. 2a and Supplementary Fig. 6a–d). By contrast, incubation of AauA with D-ribulose resulted in a dramatic increase in $T_m$ of up to 16 °C. To verify and quantify these findings, we used Isothermal Titration Calorimetry (ITC) to gain insight into the thermodynamic binding properties of AauA. Titration of L-arabinose into AauA showed a weak binding affinity ($K_D$) of 598 ± 132 μM for the sugar, whereas D-ribulose was attributable to a >200-fold greater affinity for the protein ($K_D$ = 1.98 ± 0.3 μM), indicating it is the likely preferred substrate of the system (Fig. 2b, c). To investigate the mechanistic basis of these distinctions in substrate affinity, we determined the AauA crystal structure bound to D-ribulose (Fig. 2d and Supplementary Fig. 7). The structure of AauA revealed a two-domain architecture, which is typical of other periplasmic SBPs. At the interface of the two domains, additional electron density became evident during refinement, allowing us to unambiguously model alpha-D-ribulose within the binding pocket (Supplementary Fig. 8a). The sugar is sandwiched between Trp43 and His167. The ring of alpha-D-ribulose stacks against the indole group of Trp43, and His167 forms hydrogen bonds with the hydroxyl groups at C2 and C4 positions. Additional hydrogen bonds are formed between Glu118 and C2 and C3 hydroxyls, Lys37 and C3 hydroxyl, Glu196 and Ser223 with hydroxyl group at C5 position (Fig. 2e). ConSurf analysis of AauA identified the conserved nature of the coordinating residues within the binding site (Supplementary Fig. 8b). To qualitatively compare the binding poses of D-ribulose versus L-arabinose,

we used the crystal structure of AauA to perform molecular docking of L-arabinose into its binding pocket using AutoDock Vina. Six different conformations of L-arabinose were docked into the binding site, and while all conformers occupied a similar binding position as D-ribulose, the predicted binding mode of L-arabinose exhibited fewer contacts within the substrate pocket of AauA. This could explain the weaker affinity of AauA towards L-arabinose, since modelling of AauA in complex with this sugar suggests that it would form fewer hydrogen bonds in any configuration over D-ribulose (Supplementary Fig. 8c). Taken together, these data offer a mechanistic explanation as to why our initial studies on Aau failed to identify a strong fitness defect associated with growing a deletion mutant on L-arabinose and suggest that D-ribulose may in fact be the system's preferred substrate in vivo.

## Aau and Rbl facilitate D-ribulose transport in both EHEC and *C. rodentium*

Given that *C. rodentium* appeared to encode a dedicated D-ribulose utilisation system and the higher affinity of AauA for D-ribulose, we next wanted to investigate if Aau played a role in EHEC fitness during growth on this sugar (Fig. 3a). However, our first observation was that wild type EHEC was unable to grow on D-ribulose as a sole carbon source (Fig. 3b). We reasoned that this was likely because the Aau locus did not encode an associated D-ribulokinase (such as RblK of the Rbl system, which is essential for growth of *C. rodentium* on D-ribulose). To test this, we constitutively expressed the entire *C. rodentium* Rbl system from a plasmid (pSU-*rbl*) in EHEC and found that this permitted

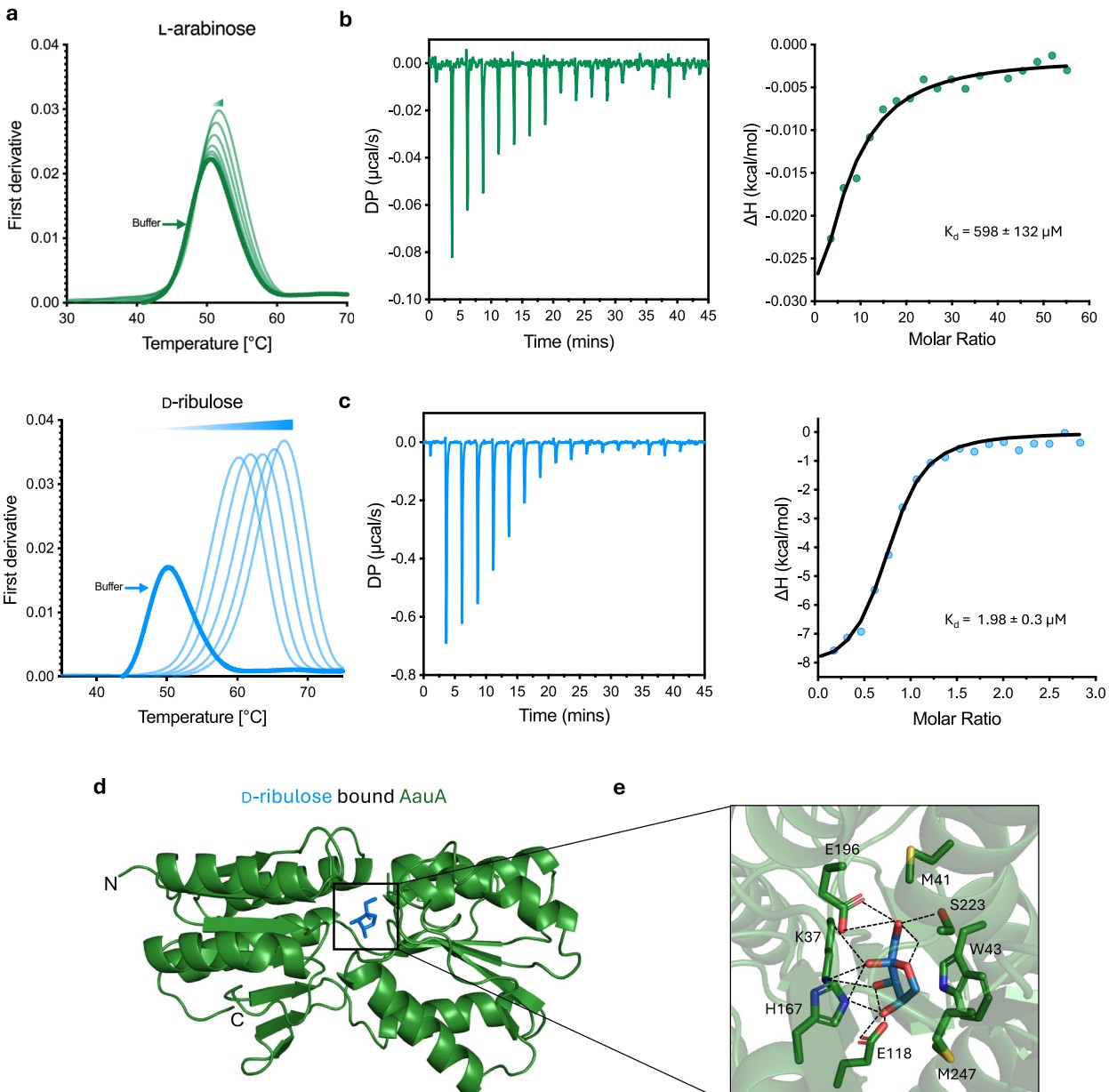

**Fig. 2 | The AauA solute binding protein of EHEC displays higher affinity for ᴅ-ribulose over ʟ-arabinose. a**, NanoDSF data depicting the shift in melting temperature (ΔT_m) of purified AauA in the presence of decreasing concentrations (2-fold dilutions from 7.5 mM to 0.03 mM (Right to left)) of ʟ-arabinose (green) or ᴅ-ribulose (blue). The buffer only control is illustrated in bold. NanoDSF melting experiments were performed in technical triplicate ($n = 3$). **b**, Representative ITC thermogram of ʟ-arabinose titrated into purified AauA. The integration of heats derivative curve is illustrated on the right, with the corresponding calculated $K_d$ shown. **c** Equivalent ITC data quantifying the binding of AauA with ᴅ-ribulose. **d** X-ray crystal structure (Green) of AauA bound with ᴅ-ribulose (Blue) at the binding cleft. The N- and C-termini are indicated. **e** Detailed view of the binding cleft and the interactions formed between the residues of AauA with ᴅ-ribulose shown as sticks. Hydrogen bonds are represented as black dashed lines. Source data are provided as a Source Data file.

growth on ᴅ-ribulose as a source of nutrition (Fig. 3c). To address whether Aau was capable of transporting ᴅ-ribulose in an alternative manner, we constitutively expressed Aau from a plasmid (pSU-*aau*) transformed into the *C. rodentium* Δ*rblABCD* background. The rationale behind this experiment was that Δ*rblABCD* was deleted for the Rbl transporter genes but still encoded the ᴅ-ribulokinase (*rblK*) necessary for growth on ᴅ-ribulose (Fig. 3d, e). Expression of Aau in this genetic background restored the mutant's ability to grow on ᴅ-ribulose, comparably to that of wild type *C. rodentium* (Growth constant (k) = 0.085 h⁻¹ vs. 0.082 h⁻¹, *P* = 0.4937) (Fig. 3f). Taken together, these data indicate that both Aau and Rbl transport ᴅ-ribulose into the cell,

but that EHEC lacks a dedicated ᴅ-ribulokinase required for metabolism of this sugar.

## EHEC hijacks the canonical ʟ-arabinose utilisation machinery to facilitate metabolism of ᴅ-ribulose

We next wanted to understand why AauA displayed higher affinity for ᴅ-ribulose, when EHEC seemingly cannot grow on this sugar as a sole carbon source. Aau expression is induced by ʟ-arabinose, which is dependent on the transcription factor, AraC[17,24]. To test whether ᴅ-ribulose could regulate the Aau system, we grew EHEC carrying the reporter plasmid pMK1*lux*-P_*aau* in MEM-HEPES supplemented with

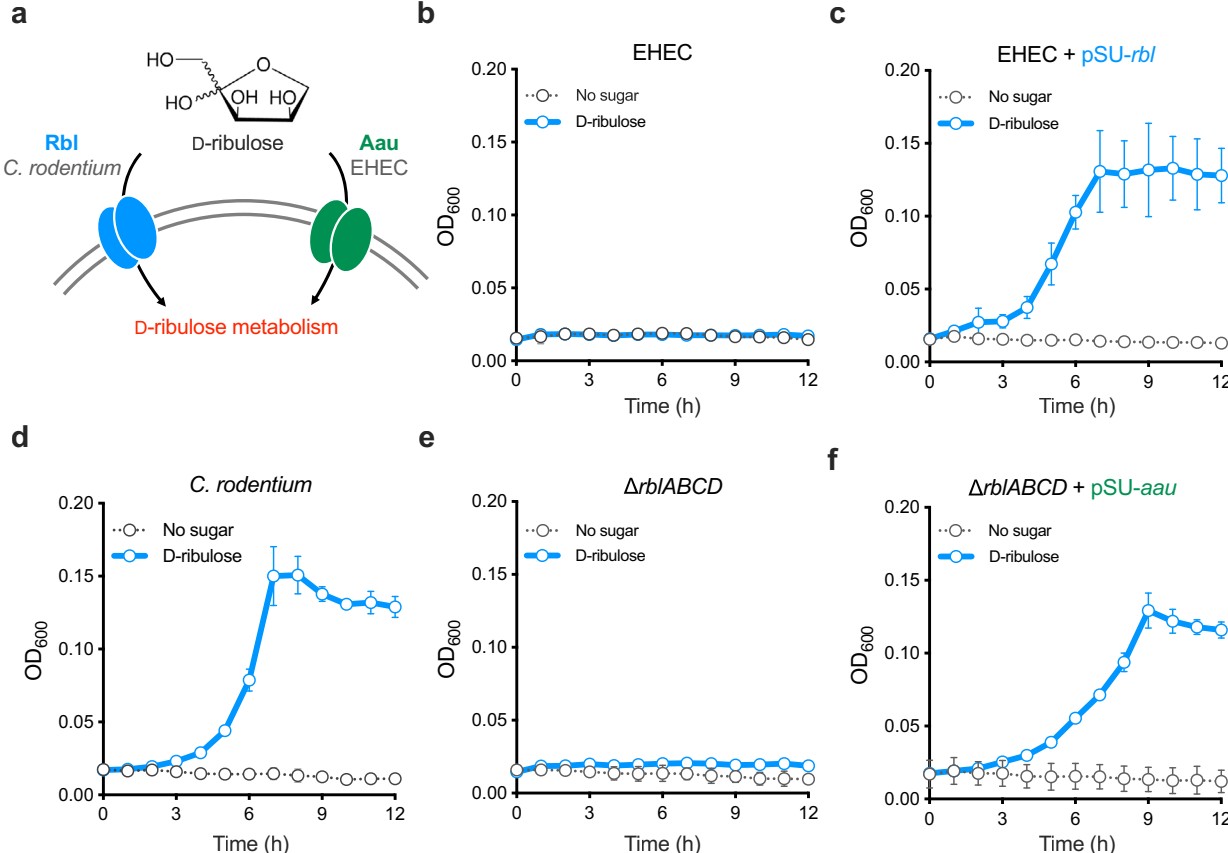

**Fig. 3 | Aau and Rbl mediate D-ribulose transport in both EHEC and *C. rodentium*. a** Schematic illustration of the similar roles for Aau and Rbl in D-ribulose transport in both species. **b** Growth analysis of wild type EHEC in M9 minimal media supplemented with D-ribulose as a sole carbon source. **c** Growth analysis of EHEC Δ*aau* cross-complemented with the entire Rbl system (pSU-*rbl*) from *C. rodentium* in M9 plus D-ribulose. **d** Growth analysis of wild type *C. rodentium* in M9 plus D-ribulose. **e** Growth analysis of *C. rodentium* Δ*rblABCD* (*ROD_24811-41*; ABC transporter components only) in M9 plus D-ribulose. **f** Growth analysis of *C. rodentium* Δ*rblABCD* (*ROD_24811-41*) in M9 plus D-ribulose cross-complemented with the Aau system (pSU-*aau*) from EHEC. For all experiments illustrated in **b**, **f**, M9 minimal media were supplemented with 0.5 mg/mL D-ribulose (blue). The no sugar control (black) indicates the respective strain inoculated into M9 without a carbon source. All growth experiments were performed on three independent occasions (*n* = 3), and the error bars represent standard deviation from the mean. Source data are provided as a Source Data file.

either L-arabinose or D-ribulose. MEM-HEPES contains D-glucose and amino acids, thus allowing us to monitor expression from pMK1*lux*-P*aau* without relying on the ability to grow on L-arabinose or D-ribulose as a sole carbon source. As previously observed, L-arabinose strongly induced *aau* transcription. However, despite the dramatically higher affinity of AauA for D-ribulose as a substrate, Aau remained transcriptionally silent in response to this sugar (Fig. 4a). We therefore reasoned that, due to AraC being essential for Aau expression, activation of the transporter in response to L-arabinose would likely allow uptake of D-ribulose, and potentially its catabolism via an alternative pathway. To test this, we grew EHEC in M9 minimal media supplemented with L-arabinose at a concentration sufficient to trigger Aau expression (0.1 mg/mL) in addition to D-ribulose (Fig. 4b). The combination of both sugars ($k = 0.067\,h^{-1}$) supported significantly enhanced growth over that generated by L-arabinose alone ($k = 0.033\,h^{-1}$; $P = 0.004$). Importantly, growth of the Δ*aau* strain under these conditions reversed the fitness advantage elicited by D-ribulose ($k = 0.03\,h^{-1}$ vs. $0.044\,h^{-1}$; $P = 0.123$). (Fig. 4c). These data suggest that EHEC can in fact grow on D-ribulose as a source of nutrition, but only in the presence of L-arabinose, which is essential for triggering *aau* expression.

The absence of an associated metabolic enzyme at the *aau* locus suggested that an additional factor, dependent on L-arabinose, is required for the utilisation of D-ribulose in EHEC. Given that the *rbl* locus of *C. rodentium* encodes a dedicated D-ribulokinase (RblK) that is essential for the metabolism of this sugar, we predicted that an analogous mechanism is likely at play in EHEC via an orphan kinase encoded elsewhere on the chromosome. BLASTP searches for *rblK* homologues in the prototype EHEC strains, ZAP193 and EDL933, failed to identify any predicted D-ribulokinase. However, EHEC encodes the L-ribulokinase AraB, a component of the canonical L-arabinose utilisation operon (*araBAD*). L-ribulokinase enzymes have been reported to be more promiscuous in terms of sugar specificity than their D-ribulokinase counterparts[25,26]. Furthermore, expression of AraB is dependent on AraC for its regulation, similarly to Aau. We therefore reasoned that AraB activity may account for the processing of D-ribulose into cellular metabolism. Whilst sharing only 28% identity at the amino acid level, overlaying the X-ray crystal structure of AraB from the Protein Data Bank (3QDK) with the predicted structure of RblK revealed strong conservation in the domain architecture between both proteins (Fig. 5a). Furthermore, all but one residue (A96) involved in L-ribulose binding by AraB were conserved in RblK and lay within a region likely to comprise the substrate-binding cleft (Fig. 5b and Supplementary Fig. 9). Guided by these striking similarities, and the need for L-arabinose to induce the transcription of *aau* and *araBAD*, we hypothesised that AraB was likely the additional missing factor and substitutionally phosphorylates D-ribulose, therefore allowing for utilisation of this nutrient.

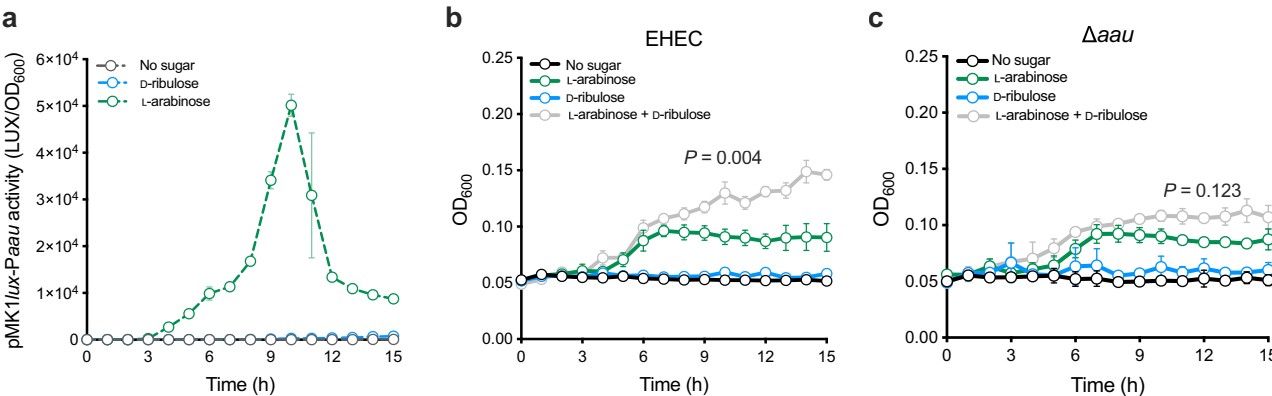

**Fig. 4 | EHEC requires ʟ-arabinose to trigger uptake and utilisation of ᴅ-ribulose. a** Transcriptional reporter assay of EHEC transformed with the pMK1*lux*-P*aau* plasmid cultured in MEM-HEPES alone or supplemented with 0.5 mg/mL ᴅ-ribulose (blue) or ʟ-arabinose (green). Data are depicted as luminescence units (LUX) divided by optical density (OD₆₀₀) of the culture at each timepoint. **b** Growth analysis of wild type EHEC in M9 minimal media supplemented with 0.5 mg/mL ᴅ-ribulose, 0.1 mg/mL ʟ-arabinose or a mixture of both sugars (grey).

**c** Comparative growth analysis of Δ*aau* performed in the exact same conditions as (**b**). Error bars for reporter assays and growth curves represent the standard deviation from mean of three independent experiments (*n* = 3 biological replicates). Non-linear regression was used to fit growth models, from which growth rate constants **k** were extracted and compared using a two-tailed student's *t*-test. Source data are provided as a Source Data file.

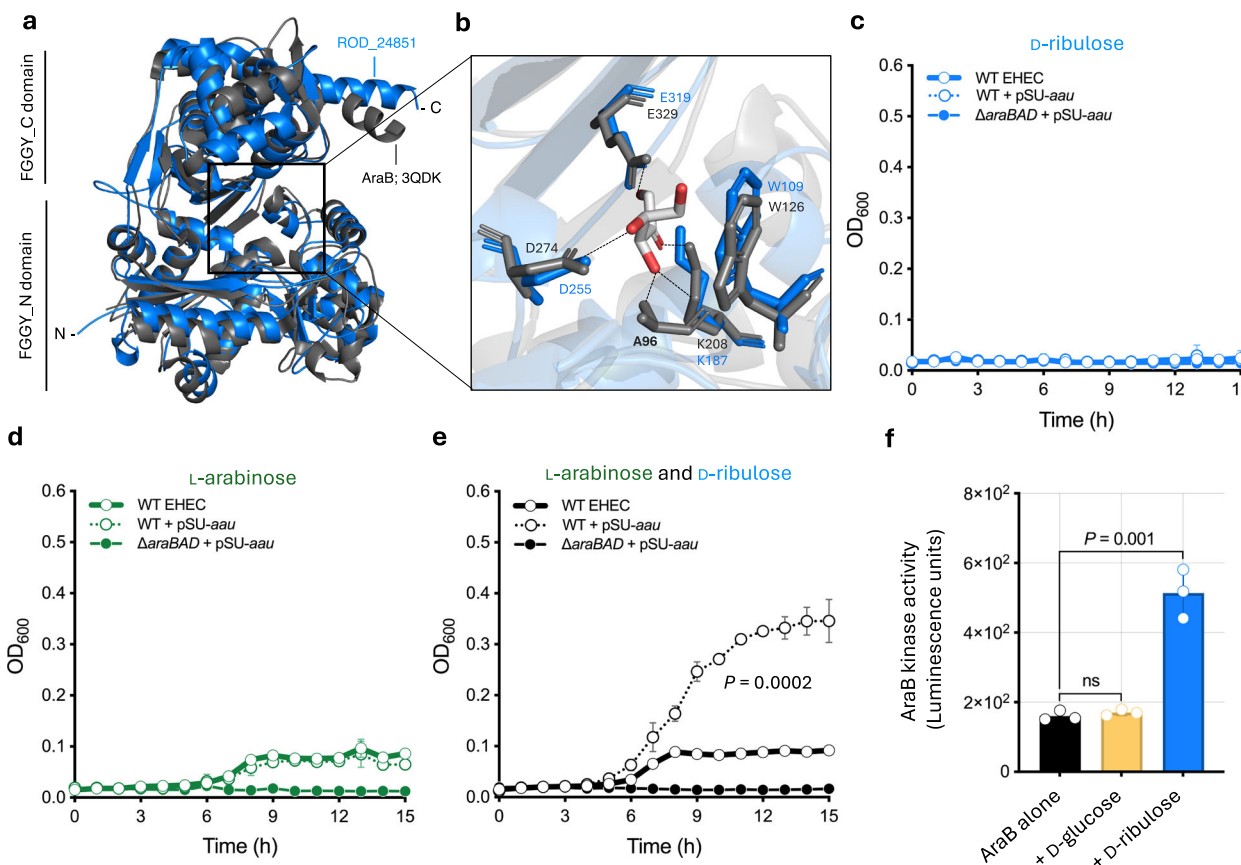

**Fig. 5 | ᴅ-ribulokinase activity of AraB is essential for growth of EHEC on ᴅ-ribulose. a** Overlayed AlphaFold3 model of RblK (Blue) from *C. rodentium* with the crystal structure of AraB from the Protein Data Bank (3QDK) (Grey). **b** Detailed view of the binding cleft from the overlay in panel, with ʟ-ribulose included to illustrate the key interacting residues known for AraB. The dashed lines indicate hydrogen bonds. **a** illustrating conservation of key residues in RblK with the known FGGY-family carbohydrate kinase, AraB. **c** Growth analysis of wild type EHEC (str. TUV93-0; encoding a non-functional Aau system), EHEC + pSU-*aau*, and Δ*araBAD* + pSU-*aau* in M9 minimal media supplemented with ᴅ-ribulose as a sole carbon source. **d**, Growth analysis of EHEC, EHEC + pSU-*aau*, and Δ*araBAD* + pSU-*aau* in M9 supplemented with ʟ-arabinose as a sole carbon source. **e** Growth analysis of

EHEC, EHEC + pSU-*aau*, and Δ*araBAD* + pSU-*aau* in M9 supplemented with ʟ-arabinose and ᴅ-ribulose as carbon sources. Error bars for growth curves represent the standard deviation from three independent experiments (*n* = 3 biological replicates). Non-linear regression was used to fit growth models, from which growth rate constants k were extracted and compared using a two-tailed student's *t*-test. **f** Kinase assay using purified recombinant AraB from EHEC (str. ZAP193) incubated with buffer, ᴅ-glucose or ᴅ-ribulose. Kinase activity was measured as raw luminescence units. The error bars represent the standard deviation from the mean of three independent experiments (*n* = 3 biological replicates). Statistical significance was determined using a two-tailed Student's *t*-test; ns indicates not significant. Source data are provided as a Source Data file.

To dissect this potential mechanism, we next constitutively expressed plasmid-derived Aau in the EHEC TUV93-0 background (an EDL933 derivative strain that naturally encodes a non-functional Aau system), thereby removing the requirement for L-arabinose as an inducer of *aau*. As expected, no growth was observed on D-ribulose as a sole carbon source and constitutive expression of Aau did not affect this result (Fig. 5c). Conversely, culture of these strains on L-arabinose supported growth but constitutive Aau expression provided no additional advantage (Fig. 5d). However, constitutive expression of Aau in the presence of both L-arabinose and D-ribulose ($k = 0.15\,\mathrm{h}^{-1}$) resulted in a significant fitness advantage over that observed from growth on L-arabinose alone ($k = 0.092\,\mathrm{h}^{-1}$; $P = 0.0002$) (Fig. 5e). Furthermore, this growth advantage was completely abolished in the Δ*araBAD* background, demonstrating that D-ribulose utilisation by EHEC is dependent on both Aau and AraBAD. To verify that this phenotype was indeed mediated by the kinase activity of AraB, we overexpressed and purified a recombinant His-tagged version of the enzyme from EHEC str. ZAP193 (Supplementary Fig. 10a–c). AraB activity was determined using a luciferase-based reporter assay whereby the luminescence detected was proportional to the amount of ADP generated following ATP hydrolysis by the enzyme, and therefore kinase activity. No significant AraB activity ($P = 0.7058$) was detected when assayed against D-glucose as a control. However, when assayed against D-ribulose, AraB activity significantly increased ($P = 0.001$) over the buffer-only control (Fig. 5f). Taken together, these data conclusively indicate that L-arabinose-dependent Aau activity allows high-affinity uptake of D-ribulose and that its subsequent processing is dependent on the canonical L-ribulokinase AraB. This, to our knowledge, represents the first complete mechanism of D-ribulose uptake and utilisation described for the *E. coli* species.

### Large-scale absence of *rbl* across the *E. coli* species suggests convergent evolutionary pathways towards D-ribulose utilisation

We previously determined that *aau* gene carriage was widespread across the six major phylogenetic clades of *E. coli*, with significant enrichment being found in phylogroups E, B1, and D, and partial carriage in B2, largely comprised of extraintestinal pathogenic isolates[17]. Given the clear distinctions between *C. rodentium* Rbl and *E. coli* Aau in terms of their regulation, genomic context, and associated mechanisms, we re-analysed the same 949 *E. coli* genomes for the presence of *rbl*, finding no instances of this locus within our dataset (Fig. 6a, b). This is in sharp contrast to the widespread carriage of *aau* and ubiquitous presence of the canonical L-arabinose utilisation genes in the *E. coli* core genome[17]. To identify if D-ribulokinase encoding genes are associated with other genomic contexts, we further interrogated the Integrated Microbial Genomes database and identified evidence of FGGY-family D-ribulokinase carriage in *Escherichia* spp. Analysis of these hits revealed three clades carrying potential D-ribulokinases, each associated with distinct genomic contexts (Supplementary Fig. 11a). The largest clade (50/96 strains) identified a D-ribulokinase (previously defined as DarK, required for D-arabinose metabolism) located as an insertion within the L-fucose utilisation locus[27]. The second clade identified was comprised of strains where the predicted D-ribulokinase was associated with a ribitol 2-dehydrogenase and polyol permease, reflective of a similar ribitol utilisation locus previously reported in *Klebsiella* spp[28,29]. Lastly, in clade three, we identified 5 genomes where a predicted D-ribulokinase was associated with an ABC transporter, sharing high sequence identity and context with the Rbl system of *C. rodentium*. However, inspection of these genomes revealed that none of these isolates could be reliably classified as *E. coli*, and were instead comprised of *Enterobacter aerogenes*, *Pseudescherichia vulneris*, *Citrobacter freundii*, and *Escherichia alba* isolates (Supplementary Fig. 11b). Based on this, we can conclude that the *rbl* locus is not widely encoded by the *E. coli* species. These findings

suggest that while a dedicated D-ribulose utilisation system may be a common feature of other pathogenic species, such as those discussed above, the major mechanism of D-ribulose utilisation in *E. coli* is via the combined action of Aau and the L-arabinose utilisation machinery, as we have demonstrated here for EHEC. Therefore, we hypothesise that convergent evolution has driven the emergence of these distinct D-ribulose utilisation pathways to take advantage of this previously understudied nutrient source.

## Discussion

Freter's nutrient niche hypothesis states that within complex environments, such as the gut, foreign species must be able to utilise at least one limiting nutrient more effectively than their competitors[5,30,31]. Competition for extremely limited resources in the gut is therefore a key aspect of colonisation resistance[1]. *E. coli* is restricted to utilising simple sugars that are liberated from more complex dietary polysaccharides during fermentation[5,32]. To take advantage of this, *E. coli* employs several strategies, such as encoding numerous genetic systems for scavenging simple sugars and metabolising multiple nutrient sources simultaneously. Moreover, metabolic flexibility is a key trait of *E. coli* pathotypes, which are capable of utilising certain nutrients more efficiently than their commensal counterparts[31,32]. Therefore, the ability of *E. coli* pathotypes to effectively scavenge limited nutrients is central to overcoming colonisation resistance and establishing a foothold in the host.

We recently reported the discovery of Aau, an ABC transporter encoded on the OI-17 pathogenicity island in EHEC that is highly expressed in bovine intestinal contents and during human infection[10,17]. Aau is regulated exclusively in response to L-arabinose, a monosaccharide that is liberated from complex dietary fibre and enriched in mammalian colonic content[33,34]. Using the streptomycin-treated mouse model, wild-type EHEC displayed a strong competitive advantage (~2 orders of magnitude) over the Δ*aau* mutant[17]. While this suggested an important role for Aau in vivo, there remained ambiguity over its precise function, considering that the system only conferred an in vitro fitness advantage when overexpressed in a strain deleted for the canonical L-arabinose uptake machinery. This suggested one of two things—that Aau was only highly effective at scavenging L-arabinose within the context of the gut or that the system may have a broader substrate specificity. Here, we discovered that Aau displays ~200-fold higher affinity for D-ribulose, over L-arabinose. Furthermore, we show that L-arabinose is required for growth of EHEC on D-ribulose, as it is essential for activating *aau* transcription. Lastly, we provide a complete mechanism for D-ribulose utilisation in EHEC, by demonstrating that the L-ribulokinase AraB (also transcriptionally controlled by L-arabinose) is absolutely required for its catabolism. These results collectively provide mechanistic reasoning underlying how and why Aau provides EHEC with such a strong fitness advantage during colonisation of the mammalian gut.

D-ribulose is a ketopentose monosaccharide with roles as an intermediate metabolite in the pentose phosphate pathway and glucuronate interconversions across eukaryotes and prokaryotes. Several groups have reported the detection of D-ribulose in the murine faecal metabolome[35]. One study most strikingly identified an increased abundance of D-ribulose in the faeces of Winnie mice, a mouse model of chronic intestinal inflammation driven by a mutation in the *Muc2* gene[36]. Based on this link, we hypothesise that EHEC could take advantage of increased intestinal D-ribulose during infection, as a result of virulence factor-induced inflammation[37]. Increased availability of D-ribulose would, in turn, enhance type 3 secretion system expression via its catabolism, providing a fitness benefit in parallel with enhanced colonisation potential[17,18]. L-arabinose is a dietary pentose that is abundant in plant cell wall polysaccharides, but the precise source of free D-ribulose in the gut is not clear. However, D-ribulose has been detected in human faeces by faecal metabolome profiling[38].

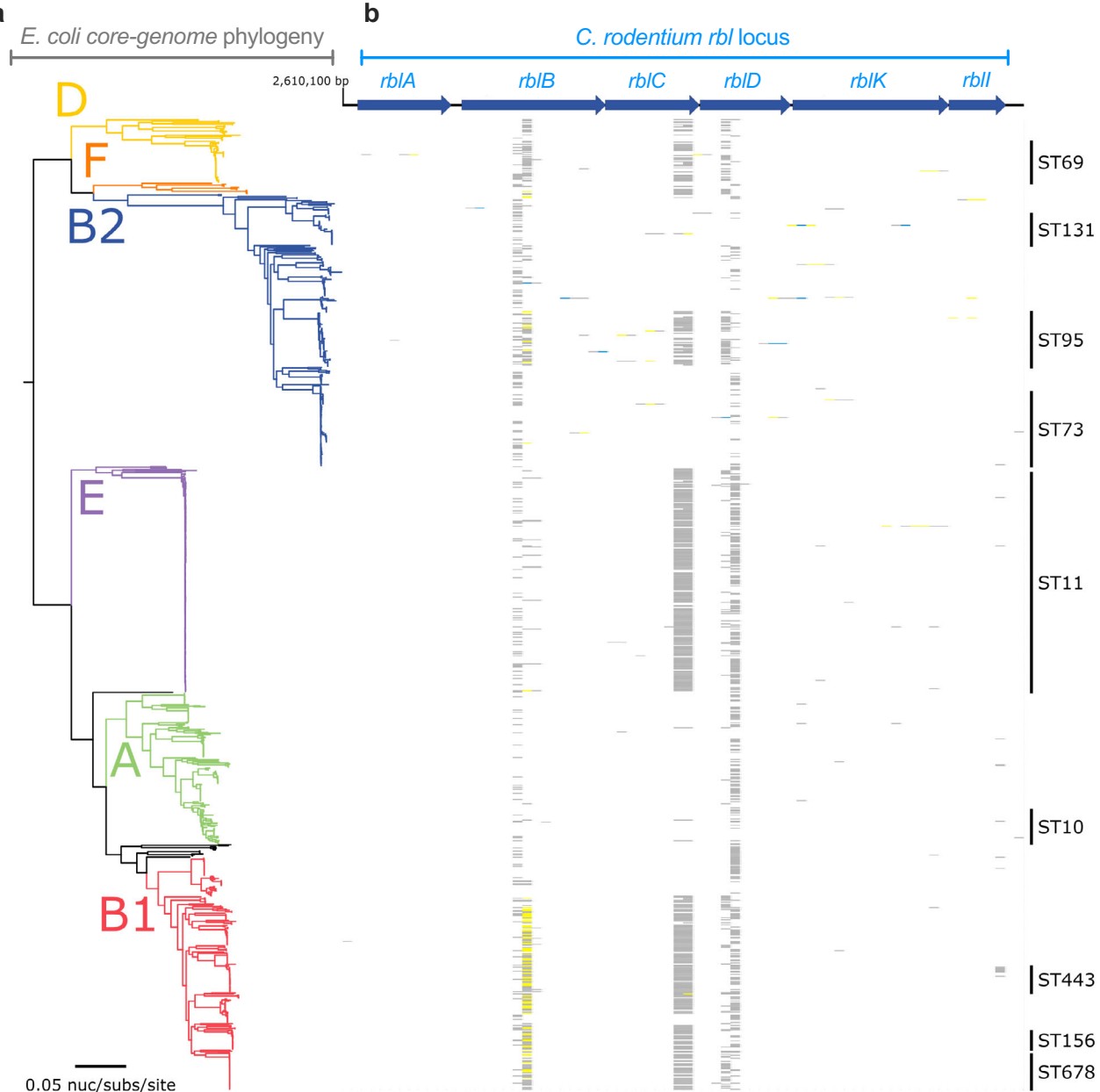

**Fig. 6 | The Rbl system from *C. rodentium* is absent from the *E. coli* species.**
**a** Maximum likelihood tree depicting the core-genome phylogeny of 949 *E. coli* isolates built from 245,518 core-genome single-nucleotide polymorphisms, called against str. EDL933 is a reference chromosome. Phylogeny is rooted according to the actual root by *Escherichia fergusonii* ATCC 35469 (omitted for visualisation). The six main phylogroups (A, B1, B2, D, E, and F) are illustrated as coloured branches. Branch lengths and scale represent the number of nucleotide substitutions per site. **b** Presence/absence of the *rbl* locus (*ROD_24811-61* from *C. rodentium* str. ICC168) is based on uniform nucleotide coverage at each 100 bp window size. The heat map illustrates coverage as ≥80% in blue, ≥50% in yellow, and ≥1% in grey. The clustering of isolate sequence types (ST) is indicated on the right. Genome accession numbers are provided as a Source Data file.

These insights collectively highlight the significance of our findings, not only for understanding EHECs association with animals but also as a pathogen of humans.

In parallel to our mechanistic characterisation of Aau, we have also discovered that the Rbl system of *C. rodentium* specifically transports ᴅ-ribulose and utilises it via a dedicated ᴅ-ribulokinase. While this system is evolutionarily distinct from Aau, it is similarly upregulated by *C. rodentium* during colonisation of the murine gut (its natural host) and essential for ᴅ-ribulose metabolism. While not widely reported in bacteria, ᴅ-ribulokinase activity has been documented in *Klebsiella* spp. and *E. coli*, as part of the ribitol and ʟ-fucose utilisation loci respectively[25,27–29]. However, our discovery is unique in that it represents a dedicated ABC transport system that transcriptionally

responds to, and is essential for, uptake and metabolism of ᴅ-ribulose. Taken together, our mechanistic dissection of Aau and Rbl function suggest convergent evolution of discrete pathways towards ᴅ-ribulose utilisation in attaching and effacing pathogens with distinct host ranges.

Cross-regulation of monosaccharide transport systems has been previously reported, typically as a means of controlling a cellular nutrient utilisation preference or hierarchy[39–42]. For example, genes of the ᴅ-xylose transport system can be directly repressed by the ʟ-arabinose-specific regulator AraC[43]. Regulation of the Aau system by AraC reflects a coordinated response that allows for the uptake and metabolism of both ᴅ-ribulose and ʟ-arabinose in vivo. While the Rbl system of *C. rodentium* is essential for utilisation of ᴅ-ribulose

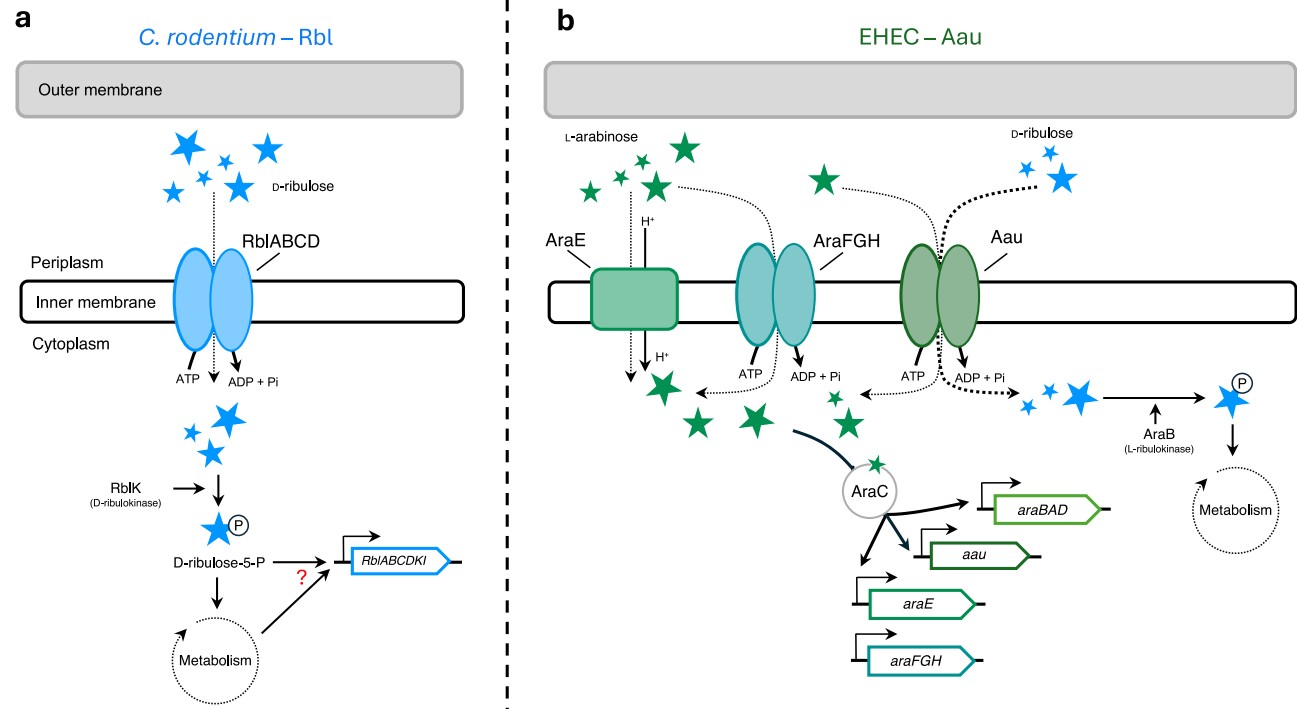

**Fig. 7 | Two distinct pathways for D-ribulose utilisation in attaching and effacing pathogens.** Schematic model illustrating; **a** the Rbl system of *C. rodentium* that is specifically regulated in response to D-ribulose, required for its uptake (via the RblABCD ABC transporter) and essential for its utilisation as a carbon source (via the RblK D-ribulokinase). The regulatory effector activating the system is unknown but is dependent on D-ribulose metabolism. **b** the Aau system of EHEC that is regulated exclusively in response to L-arabinose and mediates low affinity transport (dashed arrow) but displays dramatically higher affinity (bold dashed arrow) for D-ribulose. EHEC cannot utilise D-ribulose as a sole carbon source. However, in the presence of L-arabinose, AraC activates expression of Aau and the canonical L-arabinose utilisation machinery, leading to D-ribulose uptake (via the AauABCD ABC transporter) and catabolism (via the D-ribulokinase activity of AraB).

exclusively, it does not encode a cognate transcription factor. This suggests that cross-regulation by an alternative system (similar to Aau) may mediate the response to D-ribulose. Indeed, we did observe a weak transcriptional response of *rbl* to D-ribose (Fig. 1d), leading us to hypothesise that the D-ribose-specific transcription factor RbsR may control the *rbl* locus[44]. However, deletion of *rbsR* failed to eliminate the *C. rodentium* response to D-ribulose, leaving the mechanism of its activation unknown (Supplementary Fig. 12a). Interestingly, we also observed that transcriptional activation of *rbl* in response to D-ribulose was eliminated in the D-ribulokinase deletion background (Supplementary Fig. 12b). This suggests that D-ribulose metabolism, rather than sensing of the sugar itself, acts as the trigger for Rbl expression. While convention often argues for the function of a cognate transcription factor to sense sugars directly (as is the case with AraC and RbsR), metabolic by-products as regulatory effectors of sugar utilisation genes have been previously observed[45]. The current data also mirrors our recent finding that metabolism of pentose sugars in EHEC and *C. rodentium* mediates pathogenesis via generation of the downstream metabolite pyruvate, which in turn upregulates the type 3 secretion system, promoting colonisation of the host epithelium[17,18]. While the precise effector for *rbl* transcriptional activation remains elusive, our data support the concept of metabolism indirectly regulating important fitness factors, which has emerged as being more complex than previously appreciated.

In summary, we have identified two unique D-ribulose scavenging pathways in distinct enteric pathogens, suggesting convergent evolution towards the utilisation of this sugar (Fig. 7). The Aau and Rbl systems of EHEC and *C. rodentium*, respectively, represent defined mechanisms of D-ribulose scavenging that both centre around the activity of ABC transporters. This complements recent advances in uropathogenic *E. coli* that identified ABC transporters as key virulence

factors of the urinary tract and suggests that an expanded capacity to scavenge nutrients similarly benefits pathogenesis in the gut ecosystem[9]. Importantly, our work highlights the complexities of how metabolic flexibility is regulated in *E. coli* to provide a specific fitness benefit by recruiting activities encoded in discrete genetic loci. This raises questions about the precise mechanism and role in nature for additional nutrient uptake systems in *E. coli* that remain to be characterised.

## Methods

### Bacterial strains and growth conditions
All bacterial strains used in this study are listed in Supplementary Table 1. Single bacterial colonies were inoculated into 5 mL LB broth supplemented with the appropriate antibiotics and cultured overnight at 37 °C with shaking at 200 rpm. Overnight cultures were washed three times in PBS to remove trace LB and used to inoculate M9 minimal media supplemented with specified carbon sources or MEM-HEPES. When required, the following antibiotics were added to the media: 100 μg/mL ampicillin, 50 μg/mL kanamycin, and 20 μg/mL chloramphenicol. All chemicals and media were purchased from Merck or Thermo Fisher Scientific.

### Generation of isogenic deletion mutants
The Lambda Red recombineering system was used to generate non-polar isogenic deletions[46]. In brief, the FRT-Kanamycin or FRT-Chloramphenicol cassettes were amplified from pKD4 or pKD3 respectively. Primers contained 50 base pair overhangs homologous to the flanking regions directly up- and downstream of the gene(s) of interest to be deleted. Parental strains harbouring pKD46 were cultured in SOB media at 30 °C, containing ampicillin and supplemented with 0.1 M L-arabinose, to an OD600 of 0.4. Cells were then washed

three times in ice-cold water and concentrated 100-fold before being electroporated with 100 ng of the PCR product generated from pKD4 or pKD3. Cells were recovered in SOC media at 37 °C before plating onto LB-agar containing the appropriate antibiotic for the selection of recombinants. Positive deletions were identified by colony PCR and verified by Sanger Sequencing (Eurofins). The resistance cassette was removed by transforming mutants with pCP20, plating on LB-agar containing ampicillin, and grown overnight at 30 °C to induce FLP recombinase activity. Colonies were re-streaked non-selectively at 42 °C to cure the clean mutants of pCP20. All the primers used in this study are listed in Supplementary Table 2.

## Plasmid construction
All the plasmids used in this study were generated by restriction enzyme cloning and are listed in Supplementary Table 3. For *luxCDABE* transcriptional reporter plasmids, promoter regions corresponding to ~300 base pairs upstream of the gene of interest were amplified by PCR, gel extracted, digested, phosphatase-treated, and ligated into pMK1*lux* using restriction sites EcoRI (5′) and BamHI (3′). For complementation and overexpression plasmids, genes/operons of interest were amplified by PCR, purified as above, and ligated into pSU-PROM using restriction sites BamHI (5′) and XbaI (3′), and pET28a using restriction sites NcoI (5′) and XhoI (3′), respectively. All restriction enzymes, Q5 high-fidelity polymerase, Antarctic Phosphatase, and T4 ligase were purchased from New England Biolabs. Cloning was confirmed by Sanger Sequencing of inserts (Eurofins).

## Microplate reader growth assays
All growth experiments were conducted in clear, flat-bottom, 96-well polystyrene microtiter plates. The $OD_{600}$ of bacterial cultures was measured continuously over 15 h using a FLUOstar Omega microplate reader (BMG Labtech). Individual culture volumes (200 μL total) were inoculated with the strain of interest at a ratio of 1:100.

## LUX-promoter fusion reporter assays
To determine promoter activity, $OD_{600}$ and absolute luminescence of cultures transformed with reporter plasmids were measured in tandem using a FLUOstar Omega microplate reader (BMG Labtech). For each time point, absolute luminescence values were divided by the $OD_{600}$ to calculate the relative luminescence units (RLU). Assays were measured in real time in white-walled/clear flat-bottom microtiter plates. Individual wells were inoculated with the strain of interest at a ratio of 1:100. All reporter assays were performed on three independent occasions.

## SDS-PAGE and immunoblot
Bacterial samples were normalised by $OD_{600}$ of the culture at the time of sampling, and the corresponding media-free pellets were boiled in 4x Laemmli buffer for 10 min. Samples were separated by SDS-PAGE using the Bio-Rad Mini-PROTEAN system running at 150 volts for 90 minutes. Gels were either stained with Coomassie or transferred to 0.45 μm nitrocellulose membrane (GE Healthcare) at 30 volts for 90 minutes. Membranes were washed in PBS-Tween and blocked with 5 % skim milk solution. Blocked membranes were then incubated with a HRP-conjugated antibody (anti-His; 1:3000) overnight at 4 °C. Immunoblots were then washed for 10 min with PBS-Tween (x3) before being exposed to the SuperSignal West Pico chemiluminescent substrate solution (Pierce) and imaged on a G:Box Chemi system (Syngene).

## Protein purification by immobilized metal affinity and size exclusion chromatography
Commercial *E. coli* BL21 (DE3) competent cells (New England Biolabs) were used to overexpress recombinant proteins from pET28a(+). In brief, cells were cultured in LB media supplemented with kanamycin at 37 °C and 200 rpm shaking until an $OD_{600}$ of 0.6-0.8 was reached. Protein expression was induced by supplementing cultures with 0.5 mM isopropyl β-d-1-thiogalactopyranoside and further culturing at 16 °C overnight with shaking. Cells were harvested by centrifugation at 4000 rpm and 4 °C for 20 min, prior to resuspension in wash buffer (20 mM Tris, 200 mM NaCl, pH 8.0) and lysis via sonication (low intensity, 0.5 s cycling; Braun Labsonic U sonicator). Lysates were clarified by centrifugation at 15,000 rpm and 4 °C for 30 min. Protein was purified from clarified lysates using immobilized metal affinity ion chromatography on cobalt TALON™ resin (Takara). Protein was further purified by size-exclusion chromatography using the Superdex S200 column (Cytiva) in 20 mM HEPES, pH 7.4, 200 mM NaCl. Protein purity was assessed using SDS-PAGE, validated by immunoblot, and concentration determined using a Nanodrop 2000. Proteins were then dialysed overnight with a molecular weight cut-off of 10,000 Da in 50 mM HEPES and 150 mM NaCl (pH 7.0) at 4 °C.

## Thermal denaturation assay using NanoDSF
The monosaccharides L-arabinose, D-ribose, D-ribulose, and D-glucose were added to purified AauA at a final concentration of 15-0.03 mM (sequential 2-fold dilutions). A buffer only condition (50 mM HEPES, 150 mM NaCl; pH 8.0) was included as a control. Samples were loaded into standard grade NanoDSF capillaries and then a Prometheus NT.48 device (Nanotemper). Samples were heated from 20 °C to 80 °C with a slope of 1 °C/min.

## Isothermal Titration calorimetry (ITC)
Monosaccharide substrates (L-arabinose, D-ribose, and D-glucose) were resuspended in the same buffer (50 mM HEPES, 150 mM NaCl, pH 7.0) as AauA following overnight dialysis at 4 °C. Monosaccharides (0.5–10 mM) were independently injected into the sample cell of a Microcal PEAQ-ITC instrument (Malvern Panalytical) containing 70 μM AauA or buffer. In total, the ligand was injected 19 times, starting with an initial injection of 0.4 μL, and then 2 μL thereafter. All titrations were carried out at 25 °C. Heat of dilution was determined by titrating monosaccharides into buffer and subtracted from all the experiments. Data were fitted to a. single-site binding model using Microcal PEAQ-ITC analysis software (v1.4).

## X-ray crystallography
Purified AauA was concentrated to 33 mg/mL, mixed with 10 mM D-ribulose and incubated for 30 min on ice prior to high-throughput crystallisation trials. Sitting-drop vapour diffusion screens were set up using the Mosquito robot (SPT Labtech) and incubated at 19 °C. Crystals appeared after a few days in the Structure screen (Molecular Dimensions). Crystallisation conditions consisted of 10 % PEG 1000, 10 % PEG 8000 (condition H10). Crystals were cryo-protected using 20 % PEG400 and cryo-cooled in liquid nitrogen.

All X-ray diffraction data were collected at Diamond Light Source (Harwell, UK). Datasets were processed using xia2 and merged and scaled in Aimless v0.8.2[47]. Molecular replacement was performed in MolRep v11 using a predicted model of AauA from AlphaFold[48]. Refinement was carried out using REFMAC v5.8.0425 with anisotropic B-factors, followed by cycles of manual building and fitting in Coot v0.9.8.93[49,50]. Electron density was analysed using Polder OMIT maps[51]. The final model was deposited in the Protein Data Bank (PDBID: 9I1M). Data collection and refinement statistics are summarised in Supplementary Table 4.

## Molecular docking
AutoDock Vina v1.2 was used to dock L-arabinose into the binding site of AauA[52]. 3D structures of 6 different conformers of L-arabinose were obtained from PubChem and were prepared and optimised using Phenix eLBOW v1.20.1[53]. The exhaustivity parameter for

docking was set to nine. Ligand and protein were prepared using AutoDock tools v1.5.7 (http://mgltools.scripps.edu/), with polar hydrogens added to the protein. The highest-scoring model was selected. The grid size for docking was: x dimension 16, y dimension 24, z dimension 16. Grid box parameters (x,y,z) were 24.87, 3.954, 2.622.

## Kinase assays

L-ribulokinase (AraB) activity was assayed against D-ribulose at 37 °C for 15 minutes using the ADP-Glo™ Kinase assay kit (Promega), following the manufacturer's guidance. Kinase reactions were set up to contain 100 μM D-ribulose, 0.5 μM AraB, 200 μM ATP, and kinase buffer (Tris pH 7.5, 20 mM $MgCl_2$, 0.1 mg/mL BSA). Control reactions whereby D-ribulose was substituted for D-glucose were also set up. ADP-Glo™ was added to the reaction and incubated at room temperature for 40 min to terminate kinase activity. Kinase detection reagent was then added and incubated at room temperature for a further 40 min, enabling ADP to be converted to ATP and introduce luciferase and luciferin to detect ATP. The luminescence output was then measured using a FLUOstar Omega microplate reader (BMG Labtech). All reactions were conducted in a white-walled/clear flat-bottom 96-well microtiter plate.

## Bioinformatics

All nucleotide and amino acid sequences used were retrieved from the National Centre for Biotechnology Information (NCBI) database. The Kyoto Encyclopaedia of Genes and Genomes (KEGG) was used to obtain details of the ROD_2481-61 (Rbl) locus. Signal peptide analyses were performed using SignalP 5.0[54]. Gene carriage analysis across 949 *E. coli* genomes was performed exactly as described previously using our existing dataset, except that read mapping was screened against the ROD_24811-61 (Rbl) locus from *C. rodentium* ICC168 genome (GenBank: FN543502)[17]. InterPro Scan was implemented to obtain functional analysis of proteins and prediction of potential domains[55]. *E. coli* strains encoding a FGGY-family carbohydrate kinase (TIGR01315) were retrieved from the Integrated Microbial Genomes and Microbiomes database[56]. For accuracy, species designations were validated by extracting samples of the respective genome sequence in question and using BLAST to assign a species designation based on the most reliably similar sequences. Redundant protein sequences were removed, with the remaining sequences were aligned using the MUSCLE algorithm[57]. This alignment was subsequently used to construct a maximum likelihood phylogenetic tree consisting of 100 bootstrap replicates, carried out in MEGA11 (v11.0.13)[58]. Metadata was obtained from IMG and Enterobase databases[56,59,60]. Structural predictions of RblK were generated using the AlphaFold3 server and visualised in PyMol (v2.6)[61,62]. For structural overlaying of ribulokinases, the structure of AraB from *Bacillus halodurans* (3QDK) was retrieved from the Protein Data Bank[63]. All images were generated using PyMol. Experimentally determined structure of AauA was submitted to ConSurf[64] web server and ran using the default parameters.

## Statistical analysis

Generation of graphs and statistical calculations was done using GraphPad Prism version 10. Where applicable, statistical analysis was done using the Student's *t*-test. *P* values of less than or equal to 0.05 were considered as statistically significant. To determine the growth rate constant ($k$) of bacterial growth curves a non-linear regression was applied across replicates prior to statistical analysis.

## Reporting summary

Further information on research design is available in the Nature Portfolio Reporting Summary linked to this article.

## Data availability

The X-ray crystal structure of AauA has been deposited to the Protein Data Bank under the accession ID 9I1M. Source data for Figs. 1–6 and Supplementary Figs. 1–6 and 10–12 are provided with this paper as a Source Data file. Publicly available genome sequences used in Fig. 6 and Supplementary Fig. 11 were obtained from the NCBI Sequence Read Archive and the Integrated Microbial Genomes and Microbiomes database, with the corresponding metadata and accession numbers provided in the Source Data file. Source data are provided with this paper.

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

## Acknowledgements

We are very grateful to Dr Nicky O'Boyle for his critical appraisal of this research and Dr David Bolam for access to the Isothermal Titration Calorimeter. We also thank Diamond Light Source for access to macromolecular crystallography beamlines (proposal mx32736-66). This work was funded by an Academy of Medical Sciences/Wellcome Trust Springboard Award [SBF005\1029], a Royal Society Research Grant [RGS\R2\202100], a Medical Research Council (UKRI) Career Development Award [MR/X007197/1] and a Newcastle University Faculty Fellowship awarded to J.P.R.C. I.J. is supported by a Royal Society University Research Fellowship [URF\R1\231752] and a Newcastle University Academic Track Fellowship. For the purpose of open access, the authors have applied a Creative Commons Attribution (CC BY) licence to any author-accepted manuscript version arising from this submission.

## Author contributions

C.C. and J.P.R.C. conceptualised and designed the research. C.C., K.B., and I.J. performed the research. C.C., K.B., R.T.W., I.J., and J.P.R.C. analysed the data. R.T.W., A.B., I.J., and J.P.R.C. contributed reagents/analytical tools. C.C., I.J., and J.P.R.C. wrote the paper with input from the other authors.

## Competing interests

The authors declare no competing interests.
