## [Transparent Peer Review file · Nature Communications]

Convergent evolution of distinct D-ribulose utilisation pathways in attaching and effacing pathogens

Corresponding Author: Dr James Connolly

Version 0:

Reviewer comments:

Reviewer #1

(Remarks to the Author)

The authors elucidated pathways for degrading D-ribulose in two enteric pathogens, *Citrobacter rodentium* and EHEC. In both organisms, the ability to transport and grow on D-ribulose provided a fitness advantage. However, the two systems operate differently. *E. coli* strains all appear to utilize Aau, an L-arabinose inducible ABC transporter and ribulokinase encoded by AraB. Hence, D-ribulose metabolism is inducible by L-arabinose. These discoveries add to known mechanisms by which enteric pathogens overcome colonization resistance by exploiting particular carbon and energy sources found in the gut.

The manuscript is well written and there are no obvious flaws. The data fully support the conclusions. The results are important because they further support competition for nutrients as a mechanism for overcoming colonization resistance.

Reviewer #2

(Remarks to the Author)

In this manuscript, Cottam and colleagues have investigated how bacteria can cope with nutrient limitation and in particular pathogenic microorganisms such as Enterohaemorrhagic *E. coli* (EHEC) or *Citrobacter rodentium*, to scavenge some sugars when competing with other species for a specific niche. They showed in particular that the L-arabinose uptake (Aau) ABC transporter from *E. coli*, whose expression is upregulated by L-arabinose, is capable to efficiently transport D-ribulose into the cell, notably because the Substrate-Binding Protein (SBP) of the transporter has a much higher affinity for this sugar ($KD \sim 2 \mu M$) than for L-arabinose ($KD \sim 600 \mu M$). Yet, the presence of L-arabinose is mandatory to trigger the D-ribulose uptake and its further utilization by *E. coli* thanks to the D-ribulokinase activity of AraB. Indeed, this latter enzyme seems to be more promiscuous than anticipated, as it was originally proposed that AraB was a L-ribulokinase enzyme as it was part of the canonical operon involved in L-arabinose utilization (araBAD). Therefore, by hijacking the L-arabinose transporter, EHEC becomes capable to use D-ribulose *in vivo*. Regarding *C. rodentium*, it has both a dedicated D-ribulose transporter and a specific enzyme to use this carbon source. Altogether, the results support a convergent evolution process that led two different pathogens to be capable to exploit the limited availability of sugar during the colonization of the gut niche.

Overall, this is a very nice paper that combines elegantly Microbiology with genetic approaches, with also Biochemical/Biophysical and even Structural Biology techniques to address the question of gut colonization upon nutrient limitation and how the pathogens cope with this situation. The paper is well-written and easy to follow even for a non-specialist and most of the conclusions are well supported by the data.

I have however one major concern about the kinase experiment. The purity of the recombinant kinase is rather limited. If the quality of the purification of the kinase cannot be increased, it would be good to have an inactive mutant that targets one of the residues in the active site as a negative control of the kinase activity. Otherwise, given the poor quality of the purification, it is presently hard to exclude that the kinase activity using D-ribulose is coming from a contaminant.

Minor corrections/typos

Line 92: "moderate identity". In fact, sequence identity is rather good (~60%) so moderate does not seem to be the appropriate term.

Lines 159-160: "As expected, addition of D-glucose resulted in no thermal shift" This result is not shown.

Line 167: "indicating it as the likely". Replace 'as' by 'is'

Lines 247-248: "in the domain architecture between both proteins (Fig. 5a). Furtherm: "ore, all but one residue (A96) involved in L-ribulose binding by AraB". It would useful to show another panel with the bound sugar.

Lines 501-502: "Protein purity was assessed using SDS-PAGE, stability determined by immunoblot and measured determined using a Nanodrop 2000. Stability determined by immunoblot ? The nanoDSF can be used to test the stability but not the immunoblot.

Lines 519-520. It is unclear to me if it was the sugar or the protein which was injected in the sample cell. I thought it was the sugar but it is said that "AauA (70 μ M) was injected 19 times" ?

Fig. 2 : what about L-ribulose ? Does it bind to the SBP. If not, it can be used as a negative control to show how specific the SBP is for D-ribulose ?

Fig. 1 : Is it possible that in EHEC, there is an orphan SBP encoded elsewhere on the chromosome and which will have a much higher affinity for L-arabinose than AauA? It could be involved in the uptake of this sugar by associating with the ABC transporter. The fact that an ABC transporter can be involved in interaction with different SBPs has been previously reported. Could the authors do a blast on the EHEC genome and check whether or not a similar SBP exists in the whole genome that might be a good candidate to play such a role?

Line 785 C is in capital letter

Line 808: "e, Detailed view of the catalytic cleft" Replace by "binding cleft" as no catalysis occurs here.

Line 813 "Schematic illustration of the complementary roles for Aau and Rbl in D-ribulose". I think 'similar roles' is more appropriate

Fig. 6: can you explain what ST69, ST131...mean ?

Supplementary Fig. 1. Panels a and b have been reversed.

Also, there are no error bars for the reporter assay, only for the growth, in contrast to what is written. Again, why not try L-ribulose as a control here?

Supplementary Fig. 2. There are no error bars in contrast to what is written.

Supplementary Fig. 3: "supplemented with 0.5 mg/mL D-ribose". Add (blue curves).

Error bars represent the standard deviation from three independent experiments (n = 3 biological replicates). Again, there are no error bars.

Jean-Michel Jault

Reviewer #3

(Remarks to the Author)

The present work is a solid combination of biochemical, structural, microbiological and molecular biology investigation, focusing mainly on the characterisation of the L-arabinose ABC transporter system (Aau) of enterohaemorrhagic *E. coli* (EHEC). It also reports a comparison of this system to the corresponding Rbl system of *C. rodentium*. In both systems a unique D-ribulose scavenging pathway has been identified, suggesting a process of "convergent evolution" towards the utilisation of this non-canonical sugar. Among other things, the study demonstrates very elegantly that EHEC can grow on D-ribulose in the presence of L-arabinose by utilizing the promiscuous activities of the L-arabinose transporter (Aau) and L-ribulokinase (AraB).

Generally speaking, the work conducted, and the results obtained, support most of the conclusions and claims reported in this paper, providing valuable information in the field of bacterial sugar-utilisation systems.

Several specific points may require further considerations:

The authors claim that the fact that EHEC can grow on D-ribulose in the presence of L-arabinose dramatically enhances the fitness of EHEC. However, there is no specific experiment in the paper supporting this particular claim. Fig 5e merely demonstrates the ability of EHEC to grow on D-ribulose in the presence of L-arabinose. Important aspects to consider in this regard are the amount of free D-ribulose in the native surroundings of the bacterium (which is probably rather low) and to what extent it actually affects fitness. In the PNAS paper of Shea et al., (reference 9), the authors state that "deletion of multiple transport systems was required to achieve substantial fitness defects in the cochallenge murine model". It seems that without specific in vivo experiments like the experiments provided by the present research group in their previous Nature

Communications paper (ref 17), the strength of the present claims may not be as high.

The phenomenon of cross-utilization has been described previously in several systems, including the utilization of lactose and galactosyl-glycerol by *G. stearothermophilus* T-1 via the cellobiose-PTS system and a bifunctional 6-phospho-b-gal/glucosidase (Shulami et al., *J. Biol. Chem.* (2020) 295(31) 10766–10780). In this respect, it is not clear that the term “convergent evolution” is appropriate in the present case.

Two additional points to consider are related to the structural characterisation of AauA, the substrate binding protein (SBP) of the Aau system. These are rather technical, and probably less critical points, yet they seem to require some further consideration, as briefly explained in the following paragraphs.

Looking at the crystallographic data listed in the supporting supplementary information (Table 4), it seems that the resolution of the crystallographic analysis has been “pushed” significantly too high. This is reflected in the completeness parameter reported, which is 79.1% for the entire data and only 18.9% for the highest resolution shell. Both of these percentages are much below the usually accepted parameters, and indicate that the actual resolution of the data is definitely not as high. Such non-justified extension of the practical resolution may often result in potentially questionable structural interpretations. Although the overall structure of AauA seems to be of high quality, the improper resolution cut of the data, and its corresponding refinement, may lead to distorted analysis of the fine details, especially in the substrate binding site. A more realistic data-cut and refinement is therefore suggested, just to be confident on the structural analysis at the protein-substrate interface and interactions.

A second point of concern is related to the comparison of the interactions of the AauA protein with a bound D-ribulose vs L-arabinose (Supplementary Fig. 7c). In such comparison it should be taken into account that the original structure determined here is that of the AauA-D-ribulose complex and that this rather flexible protein adopted a conformation that would best fit to bind this particular sugar substrate. A straightforward modelling of L-arabinose into this particular structure and conformation of AauA may therefore be a bit misleading, especially if accurate analysis of the interactions and H-bonds is desired.

Version 1:

Reviewer comments:

Reviewer #2

(Remarks to the Author)

My comments and concerns have been well addressed, particularly with the improved purity of the kinase. Please note, however, that there is a mistake in Figure S6: panels b and c have been reversed, and one trace is missing in panel d. Otherwise, this is a very nice paper. Congratulations on this impressive work.

Jean-Michel Jault

REVIEWER COMMENTS

We would like to thank all three reviewers for their very positive comments and appreciation for our work. We value the constructive feedback and have tried our best to address all the points raised. Please note any references made to Figure or line numbers correspond to the revised version of our manuscript.

Reviewer #1 (Remarks to the Author):

The authors elucidated pathways for degrading D-ribulose in two enteric pathogens, *Citrobacter rodentium* and EHEC. In both organisms, the ability to transport and grow on D-ribulose provided a fitness advantage. However, the two systems operate differently. *E. coli* strains all appear to utilize Aau, an L-arabinose inducible ABC transporter and ribulokinase encoded by AraB. Hence, D-ribulose metabolism is inducible by L-arabinose. These discoveries add to known mechanisms by which enteric pathogens overcome colonization resistance by exploiting particular carbon and energy sources found in the gut.

The manuscript is well written and there are no obvious flaws. The data fully support the conclusions. The results are important because they further support competition for nutrients as a mechanism for overcoming colonization resistance.

We thank this reviewer for their overwhelmingly positive opinion of our manuscript.

Reviewer #2 (Remarks to the Author):

In this manuscript, Cottam and colleagues have investigated how bacteria can cope with nutrient limitation and in particular pathogenic microorganisms such as Enterohaemorrhagic *E. coli* (EHEC) or *Citrobacter rodentium*, to scavenge some sugars when competing with other species for a specific niche. They showed in particular that the L-arabinose uptake (Aau) ABC transporter from *E. coli*, whose expression is upregulated by L-arabinose, is capable to efficiently transport D-ribulose into the cell, notably because the Substrate-Binding Protein (SBP) of the transporter has a much higher affinity for this sugar ($K_D \sim 2 \mu\text{M}$) than for L-arabinose ($K_D \sim 600 \mu\text{M}$). Yet, the presence of L-arabinose is mandatory to trigger the D-ribulose uptake and its further utilization by *E. coli* thanks to the D-ribulokinase activity of AraB. Indeed, this latter enzyme seems to be more promiscuous than anticipated, as it was originally proposed that AraB was a L-ribulokinase enzyme as it was part of the canonical operon involved in L-arabinose utilization (*araBAD*). Therefore, by hijacking the L-arabinose transporter, EHEC becomes capable to use D-ribulose in vivo. Regarding *C. rodentium*, it has both a dedicated D-ribulose transporter and a specific enzyme to use this carbon source. Altogether, the results support a convergent evolution process that led two different pathogens to be capable to exploit the limited availability of sugar during the colonization of the gut niche.

Overall, this is a very nice paper that combines elegantly Microbiology with genetic approaches, with also Biochemical/Biophysical and even Structural Biology techniques to address the question of gut colonization upon nutrient limitation and how the pathogens cope with this situation. The paper is well-written and easy to follow even for a non-specialist and most of the conclusions are well supported by the data.

We thank this reviewer for their very supportive comments. We have addressed their specific queries below.

I have however one major concern about the kinase experiment. The purity of the recombinant kinase is rather limited. If the quality of the purification of the kinase cannot be increased, it would be good to have an inactive mutant that targets one of the residues in the active site as a negative control of the kinase activity. Otherwise, given the poor quality of the purification, it is presently hard to exclude that the kinase activity using D-ribulose is coming from a contaminant.

The reviewer makes a valid point and one that, in retrospect, we agree with. We have repeated this experiment using recombinant AraB further purified by size exclusion chromatography. As can be seen in the revised data, the purity of the protein has been vastly improved, and the result of the experiment is consistent with our original conclusion that AraB is mediating the kinase activity against D-ribulose. This new data has now been included in the revised version of our manuscript as Supplementary Fig. 9c.

Minor corrections/typos

Line 92: “moderate identity”. In fact, sequence identity is rather good (~60%) so moderate does not seem to be the appropriate term.

We have modified the text to reflect this observation more accurately. It now reads:

“While sharing ~60% identity in amino acid sequence...”

Lines 159-160: “As expected, addition of D-glucose resulted in no thermal shift” This result is not shown.

We apologise for this data being inadvertently excluded. It has now been added as Supplementary Fig. 6d.

Line 167: “indicating it as the likely”. Replace ‘as’ by ‘is’

This has been corrected.

Lines 247-248: “in the domain architecture between both proteins (Fig. 5a). Furtherm: “ore, all but one residue (A96) involved in L-ribulose binding by AraB”. It would useful to show another panel with the bound sugar.

As suggested, L-ribulose has now been incorporated into the model, represented as Fig. 5b and hydrogen bonds have also been highlighted by dashed lines. We have also included a sequence alignment of ROD_24851 and AraB (3QDK) as Supplementary Fig. 8 to provide additional details of the residues conserved between the two kinases.

Lines 501-502: “Protein purity was assessed using SDS-PAGE, stability determined by immunoblot and measured determined using a Nanodrop 2000. Stability determined by immunoblot ? The nanoDSF can be used to test the stability but not the immunoblot.

The phrasing of this line has now been corrected.

Lines 519-520. It is unclear to me if it was the sugar or the protein which was injected in the sample cell. I thought it was the sugar but it is said that "AauA (70 μ M) was injected 19 times" ?

We apologise for the misunderstanding. The ligand was injected 19 times into the sample cell, not AauA. This has been corrected in the text.

Fig. 2 : what about L-ribulose ? Does it bind to the SBP. If not, it can be used as a negative control to show how specific the SBP is for D-ribulose ?

Thank you for the suggestion. We did not observe any transcriptional response of our system to L-ribulose. This data has now been included as Supplementary Fig. 1c (see comment below for details). We reasoned that this likely implies specificity in uptake for the D-isomer and therefore decided to keep our story focused on D-ribulose versus L-arabinose. We could not test the possibility of Aau having some affinity for L-ribulose due to recurring issues with obtaining sufficient L-ribulose from our supplier. As such, we would like to keep the data as they are as we don't feel that this addition is necessary to alter the interpretation of our current data. Nevertheless, chiral specificity of the SBP is a valid question and one that we intend to address as part of a follow up study.

Fig. 1 : Is it possible that in EHEC, there is an orphan SBP encoded elsewhere on the chromosome and which will have a much higher affinity for L-arabinose than AauA? It could be involved in the uptake of this sugar by associating with the ABC transporter. The fact that an ABC transporter can be involved in interaction with different SBPs has been previously reported. Could the authors do a blast on the EHEC genome and check whether or not a similar SBP exists in the whole genome that might be a good candidate to play such a role?

We apologise this was not clearer. There are several known transport systems for L-arabinose in *E. coli* (which we analysed in our previous paper - reference 17), one of which is the well characterised high affinity ABC transporter AraFGH. Based on our data these likely provide the cell with sufficient capacity for L-arabinose, hence the lack of strong growth phenotype associated with the *aau* mutant grown on L-arabinose. Therefore, we focused our study on D-ribulose as a substrate given the novelty of this finding, and strong growth phenotypes associated with the mutant grown in this carbon source.

Line 785 C is in capital letter

This has been corrected.

Line 808: "e, Detailed view of the catalytic cleft" Replace by "binding cleft" as no catalysis occurs here.

This has been corrected.

Line 813 "Schematic illustration of the complementary roles for Aau and Rbl in D-ribulose". I think 'similar roles' is more appropriate

This has been corrected.

Fig. 6: can you explain what ST69, ST131...mean ?

ST refers to "Sequence Type". This has now been added to the figure legend.

Supplementary Fig. 1. Panels a and b have been reversed.

Thank you for pointing this out. We have reorganised the associated result in the text and Figure legend to reflect the correct arrangement. It now reads:

Line 109-112: "In support of this, transcriptional analysis using a *ROD_24811-61* reporter plasmid (pMK1*lux*-P₂₄₈₁₁) revealed that D-ribulose significantly (up to 15-fold; $P = 0.0074$) activated expression of this system in a concentration dependent manner (Fig. 1d; Supplementary Fig. 1a) and we found that *C. rodentium* could grow in M9 minimal media supplemented with D-ribulose at concentrations as low as 0.05 mg/ml (Supplementary Fig. 1b)."

Also, there are no error bars for the reporter assay, only for the growth, in contrast to what is written.

Thank you for pointing this out. This appears to be an issue for Supplementary Figures 1 to 3 in the original submission. The full figures including error bars were present in our original figures and SI word document, so this must be due to the PDF conversion. We have corrected all incomplete graphs in the revised SI document.

Again, why not try L-ribulose as a control here?

As mentioned above, we have now included this data as Supplementary Fig. 1c, showing that when grown in the presence of 0.1 mg/ml L-ribulose no *Prbl*-lux expression is observed. This suggests that the response to ribulose is specific to the D-isomer. Please note, that due to issues obtaining sufficient L-ribulose from our supplier, we were not able to conduct these experiments over the same concentration gradient as described for D-ribulose. The updated text now reads:

Line 114-115: "Activation of this system was found to be specific to D-ribulose, with L-ribulose failing to elicit any transcriptional response."

Supplementary Fig. 2. There are no error bars in contrast to what is written.

Please see the comment above regarding the missing error bars.

Supplementary Fig. 3: "supplemented with 0.5 mg/mL D-ribose". Add (blue curves). Error bars represent the standard deviation from three independent experiments (n = 3 biological replicates). Again, there are no error bars.

Please see the comment above regarding the missing error bars.

Jean-Michel Jault

Reviewer #3 (Remarks to the Author):

The present work is a solid combination of biochemical, structural, microbiological and molecular biology investigation, focusing mainly on the characterisation of the L-arabinose ABC transporter system (Aau) of enterohaemorrhagic *E. coli* (EHEC). It also reports a comparison of this system to the corresponding Rbl system of *C. rodentium*. In both systems a unique D-ribulose scavenging pathway has been identified, suggesting a process of “convergent evolution” towards the utilisation of this non-canonical sugar. Among other things, the study demonstrates very elegantly that EHEC can grow on D-ribulose in the presence of L-arabinose by utilizing the promiscuous activities of the L-arabinose transporter (Aau) and L-ribulokinase (AraB).

Generally speaking, the work conducted, and the results obtained, support most of the conclusions and claims reported in this paper, providing valuable information in the field of bacterial sugar-utilisation systems.

We thank this reviewer for their kind and supportive comments. We have addressed their specific queries below.

Several specific points may require further considerations:

The authors claim that the fact that EHEC can grow on D-ribulose in the presence of L-arabinose dramatically enhances the fitness of EHEC. However, there is no specific experiment in the paper supporting this particular claim. Fig 5e merely demonstrates the ability of EHEC to grow on D-ribulose in the presence of L-arabinose. Important aspects to consider in this regard are the amount of free D-ribulose in the native surroundings of the bacterium (which is probably rather low) and to what extent it actually affects fitness. In the PNAS paper of Shea et al., (reference 9), the authors state that “deletion of multiple transport systems was required to achieve substantial fitness defects in the cochallenge murine model”. It seems that without specific *in vivo* experiments like the experiments provided by the present research group in their previous Nature Communications paper (ref 17), the strength of the present claims may not be as high.

Thank you to the reviewer for highlighting these issues, which are valid. Four separate points have been raised, which for clarity we have responded to individually:

- In our opinion, Figure 5e does indeed demonstrate enhanced fitness when EHEC is grown in the presence of L-arabinose and D-ribulose, as the specific growth rate is significantly greater than that of EHEC grown on L-arabinose alone. This demonstrates, importantly, that when both sugars are present the cells achieve a faster growth rate and higher maximal culture density that is attributable to the expression of Aau (which is essential for D-ribulose uptake).
- The reviewer is correct that the amount of D-ribulose is likely low, as is the case for all monosaccharides. We were unable to determine the exact concentration of free D-ribulose *in vivo*. Nevertheless, our *C. rodentium* *in vivo*

RNA-seq data did identify that the *rbl* system (which, from the experiments detailed in Figure 1, appears to be strongly induced only by D-ribulose) is indeed one of the most highly upregulated systems in the gut compared to growth in laboratory medium, even more so than L-arabinose utilisation systems. This supports the notion that D-ribulose as a regulatory trigger and substrate is likely present in vivo, regardless of the exact concentration being unknown. In addition to our point above, this therefore underscores the capacity of *E. coli* or *C. rodentium* to encode multiple systems for utilising different, likely scarce, nutrient sources in parallel to ultimately enhance their fitness.

- The reviewer is correct that Shea et al. state that “deletion of multiple transport systems was required to achieve substantial fitness defects in the cochallenge murine model”. However, this is not directly relevant to our current story and previous paper on Aau. Firstly, this was performed in a bladder colonisation model whereas our work relates to the gut environment. Sugars will likely be even more scarce in urine so its possible that, in that environment, multiple deletions are needed to achieve a major fitness defect. Nevertheless, the authors did observe fitness defects with single deletions, they were just not as exaggerated (which would be expected). Secondly, we have indeed shown in our previous paper that deletion of Aau alone results in a significant fitness defect in a gut colonisation model highlighting that it plays an important role in this environment in its own right.
- The reviewer makes a valid point that our claims may not be as high due to a lack of further in vivo experiments. However, its unclear exactly what experiments they are suggesting would answer this question. Pulling apart the exact contribution of each substrate to the Aau system in vivo would be incredibly challenging, if not impossible given the requirement of arabinose to activate the Aau system for ribulose uptake. One could make multiple deletions in Aau and the canonical arabinose uptake machinery and test these in mice maintained on a variety of sugar supplemented diets for example but I don't see how this would change our interpretation of the data seeing as we have already shown that Aau plays a role in the gut regardless of dietary supplementation and that the aim of our current study was to provide mechanistic genetic/biochemical/structural data to back up the function of Aau. I believe that we have sufficiently achieved this here and that extensive further in vivo work would not be justified to support this and is beyond the scope of our current study.

The phenomenon of cross-utilization has been described previously in several systems, including the utilization of lactose and galactosyl-glycerol by *G. stearothermophilus* T-1 via the cellobiose-PTS system and a bifunctional 6-phospho-b-gal/glucosidase (Shulami et al., J. Biol. Chem. (2020) 295(31) 10766–10780). In this respect, it is not clear that the term “convergent evolution” is appropriate in the present case.

Thank you for pointing out this reference. It was not our intention to use the term “convergent evolution” to refer the reader to the cross-utilisation aspect of Aau substrate specify. Rather, we used the term to describe the phenomenon of two completely independent D-ribulose utilisation systems (*Rbl* and *Aau*) that have evolved in two different bacterial species (*C. rodentium* and *E. coli*) to achieve the

same common goal of growth on the sugar. We believe this aligns well to the textbook definition of convergent evolution so would like to leave this unchanged.

Two additional points to consider are related to the structural characterisation of AauA, the substrate binding protein (SBP) of the Aau system. These are rather technical, and probably less critical points, yet they seem to require some further consideration, as briefly explained in the following paragraphs.

Looking at the crystallographic data listed in the supporting supplementary information (Table 4), it seems that the resolution of the crystallographic analysis has been “pushed” significantly too high. This is reflected in the completeness parameter reported, which is 79.1% for the entire data and only 18.9% for the highest resolution shell. Both of these percentages are much below the usually accepted parameters, and indicate that the actual resolution of the data is definitely not as high. Such non-justified extension of the practical resolution may often result in potentially questionable structural interpretations. Although the overall structure of AauA seems to be of high quality, the improper resolution cut of the data, and its corresponding refinement, may lead to distorted analysis of the fine details, especially in the substrate binding site. A more realistic data-cut and refinement is therefore suggested, just to be confident on the structural analysis at the protein-substrate interface and interactions.

We thank the reviewer for raising an important observation and their concerns about the resolution cutoff in our crystallographic analysis of AauA. It is true that the completeness in the highest resolution shell is rather low (18.9), however we chose to include this data based on the CC1/2 (0.98 in highest shell) and I/σ (7.5 in highest shell) values and recommendations of using all data that carry meaningful signal (Karplus and Dietrich, 2012. *Science* 336(6084):1030-1033).

To ensure that we are not overstressing our resolution limits, we re-processed the dataset to 1.50 Å which shows much better completeness statistics (see table below) and refined our structure against the new cut-off. As can be seen in the table below, there is little difference in the quality of the newly refined structures on the statistical side. More importantly, we provide both maps at 1.35 Å and 1.50 Å to the reviewer to inspect. But there is almost no difference in quality of the maps.

We therefore conclude that the current resolution cutoff for our dataset does not lead to any questionable interpretations regarding the substrate and its binding site. We would prefer to not re-upload the re-refined 1.50 Å structure and data to the PDB and keep the current 1.35 Å dataset but instead upload an unmerged dataset along with this deposition, allowing other researchers to re-process the datasets themselves if they wish.

Maps for comparison:

Original 1.35 Å maps

Re-processed 1.5 Å maps

Reprocessed data from Table S4: Data collection and refinement statistics

	AauA dataset* (PDBID: 9I1M)	AauA lower res. dataset
Data collection		
Beamline	Diamond Light Source I03	Diamond Light Source I03
Space group	C2	C3
Cell dimensions		
a , b , c (Å)	76.38, 66.94, 57.85	76.38, 66.94, 57.85
α , β , γ (°)	90, 94.45, 90	90, 94.45, 90
Resolution (Å)	38.9-1.35 (1.37-1.35)	38.07-1.50 (1.53-1.50)
R_{pim}	0.018 (0.098)	0.017 (0.057)
R_{meas}	0.033 (0.14)	0.033 (0.087)
$I / \sigma I$	29.8 (7.5)	37 (14)
$CC_{1/2}$	0.999 (0.98)	0.999 (0.994)
Completeness (%)	79.1 (18.9)	94.9 (65)
Redundancy	6.2 (3.2)	6.5 (4.2)
Refinement		
Resolution (Å)	1.35	1.50
No. reflections	50386 (574)	44137 (1510)
$R_{\text{work}} / R_{\text{free}}$	0.126/0.143	0.14/0.16
No. atoms	5509	5509
Protein	4464	4464
Ligand/ion	74	
B-factors (Å²)		
Protein	12.7	12.6
Ligand/ion	23.26	25.8
R.m.s. deviations		
Bond lengths (Å)	0.0124	0.118
Bond angles (°)	2.022	2.018
Rotamer outliers	0.42	0

Ramachandran (%)		
Favored regions	97.22	97.57
Allowed regions	2.78	
Outliers	0	
Molprobrity score	1.66	1.34

*Values in parentheses are for highest-resolution shell.

A second point of concern is related to the comparison of the interactions of the AauA protein with a bound D-ribulose vs L-arabinose (Supplementary Fig. 7c). In such comparison it should be taken into account that the original structure determined here is that of the AauA-D-ribulose complex and that this rather flexible protein adopted a conformation that would best fit to bind this particular sugar substrate. A straightforward modelling of L-arabinose into this particular structure and conformation of AauA may therefore be a bit misleading, especially if accurate analysis of the interactions and H-bonds is desired.

The reviewer is correct in their assessment about our analyses of the interactions. We have not been successful in obtaining a high-resolution structure of AauA in the presence of L-arabinose despite extensive screening attempts. Instead, we have now performed molecular docking of a number of L-arabinose conformations (as computed by PubChem) into AauA. All placed L- arabinose molecules do fit into the binding site of AauA in a similar position as D-ribulose however, as our previous analysis showed, the number of intermolecular interactions is lower. It was not our intention to mislead the readers about the AauA conformations in the presence of L- arabinose but simply to find a way to rationalise the difference in affinity and selectivity between D-ribulose and L-arabinose. This updated analysis has now been included as Supplementary Fig. 7c, with the figure legend and methods updated accordingly. The text regarding this update now reads:

Line 188-195: "To qualitatively compare the binding poses of D-ribulose versus L-arabinose, we used the crystal structure of AauA to perform molecular docking of L-arabinose into its binding pocket using AutoDock Vina. Six different conformations of L-arabinose were docked into the binding site and while all conformers occupied a similar binding position as D-ribulose, the predicted binding mode of L-arabinose exhibited fewer contacts within the substrate pocket of AauA."

REVIEWER COMMENTS

We would like to thank all three reviewers for their very positive comments and appreciation for our work. We value the constructive feedback and have tried our best to address all the points raised. Please note any references made to Figure or line numbers correspond to the revised version of our manuscript.

Reviewer #1 (Remarks to the Author):

The authors elucidated pathways for degrading D-ribulose in two enteric pathogens, *Citrobacter rodentium* and EHEC. In both organisms, the ability to transport and grow on D-ribulose provided a fitness advantage. However, the two systems operate differently. *E. coli* strains all appear to utilize Aau, an L-arabinose inducible ABC transporter and ribulokinase encoded by AraB. Hence, D-ribulose metabolism is inducible by L-arabinose. These discoveries add to known mechanisms by which enteric pathogens overcome colonization resistance by exploiting particular carbon and energy sources found in the gut.

The manuscript is well written and there are no obvious flaws. The data fully support the conclusions. The results are important because they further support competition for nutrients as a mechanism for overcoming colonization resistance.

We thank this reviewer for their overwhelmingly positive opinion of our manuscript.

Reviewer #2 (Remarks to the Author):

In this manuscript, Cottam and colleagues have investigated how bacteria can cope with nutrient limitation and in particular pathogenic microorganisms such as Enterohaemorrhagic *E. coli* (EHEC) or *Citrobacter rodentium*, to scavenge some sugars when competing with other species for a specific niche. They showed in particular that the L-arabinose uptake (Aau) ABC transporter from *E. coli*, whose expression is upregulated by L-arabinose, is capable to efficiently transport D-ribulose into the cell, notably because the Substrate-Binding Protein (SBP) of the transporter has a much higher affinity for this sugar ($K_D \sim 2 \mu\text{M}$) than for L-arabinose ($K_D \sim 600 \mu\text{M}$). Yet, the presence of L-arabinose is mandatory to trigger the D-ribulose uptake and its further utilization by *E. coli* thanks to the D-ribulokinase activity of AraB. Indeed, this latter enzyme seems to be more promiscuous than anticipated, as it was originally proposed that AraB was a L-ribulokinase enzyme as it was part of the canonical operon involved in L-arabinose utilization (*araBAD*). Therefore, by hijacking the L-arabinose transporter, EHEC becomes capable to use D-ribulose in vivo. Regarding *C. rodentium*, it has both a dedicated D-ribulose transporter and a specific enzyme to use this carbon source. Altogether, the results support a convergent evolution process that led two different pathogens to be capable to exploit the limited availability of sugar during the colonization of the gut niche.

Overall, this is a very nice paper that combines elegantly Microbiology with genetic approaches, with also Biochemical/Biophysical and even Structural Biology techniques to address the question of gut colonization upon nutrient limitation and how the pathogens cope with this situation. The paper is well-written and easy to follow even for a non-specialist and most of the conclusions are well supported by the data.

We thank this reviewer for their very supportive comments. We have addressed their specific queries below.

I have however one major concern about the kinase experiment. The purity of the recombinant kinase is rather limited. If the quality of the purification of the kinase cannot be increased, it would be good to have an inactive mutant that targets one of the residues in the active site as a negative control of the kinase activity. Otherwise, given the poor quality of the purification, it is presently hard to exclude that the kinase activity using D-ribulose is coming from a contaminant.

The reviewer makes a valid point and one that, in retrospect, we agree with. We have repeated this experiment using recombinant AraB further purified by size exclusion chromatography. As can be seen in the revised data, the purity of the protein has been vastly improved, and the result of the experiment is consistent with our original conclusion that AraB is mediating the kinase activity against D-ribulose. This new data has now been included in the revised version of our manuscript as Supplementary Fig. 9c.

Minor corrections/typos

Line 92: “moderate identity”. In fact, sequence identity is rather good (~60%) so moderate does not seem to be the appropriate term.

We have modified the text to reflect this observation more accurately. It now reads:

“While sharing ~60% identity in amino acid sequence...”

Lines 159-160: “As expected, addition of D-glucose resulted in no thermal shift” This result is not shown.

We apologise for this data being inadvertently excluded. It has now been added as Supplementary Fig. 6d.

Line 167: “indicating it as the likely”. Replace ‘as’ by ‘is’

This has been corrected.

Lines 247-248: “in the domain architecture between both proteins (Fig. 5a). Furtherm: “ore, all but one residue (A96) involved in L-ribulose binding by AraB”. It would useful to show another panel with the bound sugar.

As suggested, L-ribulose has now been incorporated into the model, represented as Fig. 5b and hydrogen bonds have also been highlighted by dashed lines. We have also included a sequence alignment of ROD_24851 and AraB (3QDK) as Supplementary Fig. 8 to provide additional details of the residues conserved between the two kinases.

Lines 501-502: “Protein purity was assessed using SDS-PAGE, stability determined by immunoblot and measured determined using a Nanodrop 2000. Stability determined by immunoblot ? The nanoDSF can be used to test the stability but not the immunoblot.

The phrasing of this line has now been corrected.

Lines 519-520. It is unclear to me if it was the sugar or the protein which was injected in the sample cell. I thought it was the sugar but it is said that "AauA (70 μ M) was injected 19 times" ?

We apologise for the misunderstanding. The ligand was injected 19 times into the sample cell, not AauA. This has been corrected in the text.

Fig. 2 : what about L-ribulose ? Does it bind to the SBP. If not, it can be used as a negative control to show how specific the SBP is for D-ribulose ?

Thank you for the suggestion. We did not observe any transcriptional response of our system to L-ribulose. This data has now been included as Supplementary Fig. 1c (see comment below for details). We reasoned that this likely implies specificity in uptake for the D-isomer and therefore decided to keep our story focused on D-ribulose versus L-arabinose. We could not test the possibility of Aau having some affinity for L-ribulose due to recurring issues with obtaining sufficient L-ribulose from our supplier. As such, we would like to keep the data as they are as we don't feel that this addition is necessary to alter the interpretation of our current data. Nevertheless, chiral specificity of the SBP is a valid question and one that we intend to address as part of a follow up study.

Fig. 1 : Is it possible that in EHEC, there is an orphan SBP encoded elsewhere on the chromosome and which will have a much higher affinity for L-arabinose than AauA? It could be involved in the uptake of this sugar by associating with the ABC transporter. The fact that an ABC transporter can be involved in interaction with different SBPs has been previously reported. Could the authors do a blast on the EHEC genome and check whether or not a similar SBP exists in the whole genome that might be a good candidate to play such a role?

We apologise this was not clearer. There are several known transport systems for L-arabinose in *E. coli* (which we analysed in our previous paper - reference 17), one of which is the well characterised high affinity ABC transporter AraFGH. Based on our data these likely provide the cell with sufficient capacity for L-arabinose, hence the lack of strong growth phenotype associated with the *aau* mutant grown on L-arabinose. Therefore, we focused our study on D-ribulose as a substrate given the novelty of this finding, and strong growth phenotypes associated with the mutant grown in this carbon source.

Line 785 C is in capital letter

This has been corrected.

Line 808: "e, Detailed view of the catalytic cleft" Replace by "binding cleft" as no catalysis occurs here.

This has been corrected.

Line 813 "Schematic illustration of the complementary roles for Aau and Rbl in D-ribulose". I think 'similar roles' is more appropriate

This has been corrected.

Fig. 6: can you explain what ST69, ST131...mean ?

ST refers to "Sequence Type". This has now been added to the figure legend.

Supplementary Fig. 1. Panels a and b have been reversed.

Thank you for pointing this out. We have reorganised the associated result in the text and Figure legend to reflect the correct arrangement. It now reads:

Line 109-112: "In support of this, transcriptional analysis using a *ROD_24811-61* reporter plasmid (pMK1*lux*-P₂₄₈₁₁) revealed that D-ribulose significantly (up to 15-fold; $P = 0.0074$) activated expression of this system in a concentration dependent manner (Fig. 1d; Supplementary Fig. 1a) and we found that *C. rodentium* could grow in M9 minimal media supplemented with D-ribulose at concentrations as low as 0.05 mg/ml (Supplementary Fig. 1b)."

Also, there are no error bars for the reporter assay, only for the growth, in contrast to what is written.

Thank you for pointing this out. This appears to be an issue for Supplementary Figures 1 to 3 in the original submission. The full figures including error bars were present in our original figures and SI word document, so this must be due to the PDF conversion. We have corrected all incomplete graphs in the revised SI document.

Again, why not try L-ribulose as a control here?

As mentioned above, we have now included this data as Supplementary Fig. 1c, showing that when grown in the presence of 0.1 mg/ml L-ribulose no *Prbl*-*lux* expression is observed. This suggests that the response to ribulose is specific to the D-isomer. Please note, that due to issues obtaining sufficient L-ribulose from our supplier, we were not able to conduct these experiments over the same concentration gradient as described for D-ribulose. The updated text now reads:

Line 114-115: "Activation of this system was found to be specific to D-ribulose, with L-ribulose failing to elicit any transcriptional response."

Supplementary Fig. 2. There are no error bars in contrast to what is written.

Please see the comment above regarding the missing error bars.

Supplementary Fig. 3: "supplemented with 0.5 mg/mL D-ribose". Add (blue curves). Error bars represent the standard deviation from three independent experiments (n = 3 biological replicates). Again, there are no error bars.

Please see the comment above regarding the missing error bars.

Jean-Michel Jault

Reviewer #3 (Remarks to the Author):

The present work is a solid combination of biochemical, structural, microbiological and molecular biology investigation, focusing mainly on the characterisation of the L-arabinose ABC transporter system (Aau) of enterohaemorrhagic *E. coli* (EHEC). It also reports a comparison of this system to the corresponding Rbl system of *C. rodentium*. In both systems a unique D-ribulose scavenging pathway has been identified, suggesting a process of “convergent evolution” towards the utilisation of this non-canonical sugar. Among other things, the study demonstrates very elegantly that EHEC can grow on D-ribulose in the presence of L-arabinose by utilizing the promiscuous activities of the L-arabinose transporter (Aau) and L-ribulokinase (AraB).

Generally speaking, the work conducted, and the results obtained, support most of the conclusions and claims reported in this paper, providing valuable information in the field of bacterial sugar-utilisation systems.

We thank this reviewer for their kind and supportive comments. We have addressed their specific queries below.

Several specific points may require further considerations:

The authors claim that the fact that EHEC can grow on D-ribulose in the presence of L-arabinose dramatically enhances the fitness of EHEC. However, there is no specific experiment in the paper supporting this particular claim. Fig 5e merely demonstrates the ability of EHEC to grow on D-ribulose in the presence of L-arabinose. Important aspects to consider in this regard are the amount of free D-ribulose in the native surroundings of the bacterium (which is probably rather low) and to what extent it actually affects fitness. In the PNAS paper of Shea et al., (reference 9), the authors state that “deletion of multiple transport systems was required to achieve substantial fitness defects in the cochallenge murine model”. It seems that without specific *in vivo* experiments like the experiments provided by the present research group in their previous Nature Communications paper (ref 17), the strength of the present claims may not be as high.

Thank you to the reviewer for highlighting these issues, which are valid. Four separate points have been raised, which for clarity we have responded to individually:

- In our opinion, Figure 5e does indeed demonstrate enhanced fitness when EHEC is grown in the presence of L-arabinose and D-ribulose, as the specific growth rate is significantly greater than that of EHEC grown on L-arabinose alone. This demonstrates, importantly, that when both sugars are present the cells achieve a faster growth rate and higher maximal culture density that is attributable to the expression of Aau (which is essential for D-ribulose uptake).
- The reviewer is correct that the amount of D-ribulose is likely low, as is the case for all monosaccharides. We were unable to determine the exact concentration of free D-ribulose *in vivo*. Nevertheless, our *C. rodentium* *in vivo*

RNA-seq data did identify that the *rbl* system (which, from the experiments detailed in Figure 1, appears to be strongly induced only by D-ribulose) is indeed one of the most highly upregulated systems in the gut compared to growth in laboratory medium, even more so than L-arabinose utilisation systems. This supports the notion that D-ribulose as a regulatory trigger and substrate is likely present in vivo, regardless of the exact concentration being unknown. In addition to our point above, this therefore underscores the capacity of *E. coli* or *C. rodentium* to encode multiple systems for utilising different, likely scarce, nutrient sources in parallel to ultimately enhance their fitness.

- The reviewer is correct that Shea et al. state that “deletion of multiple transport systems was required to achieve substantial fitness defects in the cochallenge murine model”. However, this is not directly relevant to our current story and previous paper on Aau. Firstly, this was performed in a bladder colonisation model whereas our work relates to the gut environment. Sugars will likely be even more scarce in urine so its possible that, in that environment, multiple deletions are needed to achieve a major fitness defect. Nevertheless, the authors did observe fitness defects with single deletions, they were just not as exaggerated (which would be expected). Secondly, we have indeed shown in our previous paper that deletion of Aau alone results in a significant fitness defect in a gut colonisation model highlighting that it plays an important role in this environment in its own right.
- The reviewer makes a valid point that our claims may not be as high due to a lack of further in vivo experiments. However, its unclear exactly what experiments they are suggesting would answer this question. Pulling apart the exact contribution of each substrate to the Aau system in vivo would be incredibly challenging, if not impossible given the requirement of arabinose to activate the Aau system for ribulose uptake. One could make multiple deletions in Aau and the canonical arabinose uptake machinery and test these in mice maintained on a variety of sugar supplemented diets for example but I don't see how this would change our interpretation of the data seeing as we have already shown that Aau plays a role in the gut regardless of dietary supplementation and that the aim of our current study was to provide mechanistic genetic/biochemical/structural data to back up the function of Aau. I believe that we have sufficiently achieved this here and that extensive further in vivo work would not be justified to support this and is beyond the scope of our current study.

The phenomenon of cross-utilization has been described previously in several systems, including the utilization of lactose and galactosyl-glycerol by *G. stearothermophilus* T-1 via the cellobiose-PTS system and a bifunctional 6-phospho-b-gal/glucosidase (Shulami et al., J. Biol. Chem. (2020) 295(31) 10766–10780). In this respect, it is not clear that the term “convergent evolution” is appropriate in the present case.

Thank you for pointing out this reference. It was not our intention to use the term “convergent evolution” to refer the reader to the cross-utilisation aspect of Aau substrate specify. Rather, we used the term to describe the phenomenon of two completely independent D-ribulose utilisation systems (*Rbl* and *Aau*) that have evolved in two different bacterial species (*C. rodentium* and *E. coli*) to achieve the

same common goal of growth on the sugar. We believe this aligns well to the textbook definition of convergent evolution so would like to leave this unchanged.

Two additional points to consider are related to the structural characterisation of AauA, the substrate binding protein (SBP) of the Aau system. These are rather technical, and probably less critical points, yet they seem to require some further consideration, as briefly explained in the following paragraphs.

Looking at the crystallographic data listed in the supporting supplementary information (Table 4), it seems that the resolution of the crystallographic analysis has been “pushed” significantly too high. This is reflected in the completeness parameter reported, which is 79.1% for the entire data and only 18.9% for the highest resolution shell. Both of these percentages are much below the usually accepted parameters, and indicate that the actual resolution of the data is definitely not as high. Such non-justified extension of the practical resolution may often result in potentially questionable structural interpretations. Although the overall structure of AauA seems to be of high quality, the improper resolution cut of the data, and its corresponding refinement, may lead to distorted analysis of the fine details, especially in the substrate binding site. A more realistic data-cut and refinement is therefore suggested, just to be confident on the structural analysis at the protein-substrate interface and interactions.

We thank the reviewer for raising an important observation and their concerns about the resolution cutoff in our crystallographic analysis of AauA. It is true that the completeness in the highest resolution shell is rather low (18.9), however we chose to include this data based on the CC1/2 (0.98 in highest shell) and I/σ (7.5 in highest shell) values and recommendations of using all data that carry meaningful signal (Karplus and Dietrich, 2012. *Science* 336(6084):1030-1033).

To ensure that we are not overstressing our resolution limits, we re-processed the dataset to 1.50 Å which shows much better completeness statistics (see table below) and refined our structure against the new cut-off. As can be seen in the table below, there is little difference in the quality of the newly refined structures on the statistical side. More importantly, we provide both maps at 1.35 Å and 1.50 Å to the reviewer to inspect. But there is almost no difference in quality of the maps.

We therefore conclude that the current resolution cutoff for our dataset does not lead to any questionable interpretations regarding the substrate and its binding site. We would prefer to not re-upload the re-refined 1.50 Å structure and data to the PDB and keep the current 1.35 Å dataset but instead upload an unmerged dataset along with this deposition, allowing other researchers to re-process the datasets themselves if they wish.

Maps for comparison:

Original 1.35 Å maps

Re-processed 1.5 Å maps

Reprocessed data from Table S4: Data collection and refinement statistics

	AauA dataset* (PDBID: 9I1M)	AauA lower res. dataset
Data collection		
Beamline	Diamond Light Source I03	Diamond Light Source I03
Space group	C2	C3
Cell dimensions		
a , b , c (Å)	76.38, 66.94, 57.85	76.38, 66.94, 57.85
α , β , γ (°)	90, 94.45, 90	90, 94.45, 90
Resolution (Å)	38.9-1.35 (1.37-1.35)	38.07-1.50 (1.53-1.50)
R_{pim}	0.018 (0.098)	0.017 (0.057)
R_{meas}	0.033 (0.14)	0.033 (0.087)
$I / \sigma I$	29.8 (7.5)	37 (14)
$CC_{1/2}$	0.999 (0.98)	0.999 (0.994)
Completeness (%)	79.1 (18.9)	94.9 (65)
Redundancy	6.2 (3.2)	6.5 (4.2)
Refinement		
Resolution (Å)	1.35	1.50
No. reflections	50386 (574)	44137 (1510)
$R_{\text{work}} / R_{\text{free}}$	0.126/0.143	0.14/0.16
No. atoms	5509	5509
Protein	4464	4464
Ligand/ion	74	
B-factors (Å²)		
Protein	12.7	12.6
Ligand/ion	23.26	25.8
R.m.s. deviations		
Bond lengths (Å)	0.0124	0.118
Bond angles (°)	2.022	2.018
Rotamer outliers	0.42	0

Ramachandran (%)		
Favored regions	97.22	97.57
Allowed regions	2.78	
Outliers	0	
Molprobability score	1.66	1.34

*Values in parentheses are for highest-resolution shell.

A second point of concern is related to the comparison of the interactions of the AauA protein with a bound D-ribulose vs L-arabinose (Supplementary Fig. 7c). In such comparison it should be taken into account that the original structure determined here is that of the AauA-D-ribulose complex and that this rather flexible protein adopted a conformation that would best fit to bind this particular sugar substrate. A straightforward modelling of L-arabinose into this particular structure and conformation of AauA may therefore be a bit misleading, especially if accurate analysis of the interactions and H-bonds is desired.

The reviewer is correct in their assessment about our analyses of the interactions. We have not been successful in obtaining a high-resolution structure of AauA in the presence of L-arabinose despite extensive screening attempts. Instead, we have now performed molecular docking of a number of L-arabinose conformations (as computed by PubChem) into AauA. All placed L-arabinose molecules do fit into the binding site of AauA in a similar position as D-ribulose however, as our previous analysis showed, the number of intermolecular interactions is lower. It was not our intention to mislead the readers about the AauA conformations in the presence of L-arabinose but simply to find a way to rationalise the difference in affinity and selectivity between D-ribulose and L-arabinose. This updated analysis has now been included as Supplementary Fig. 7c, with the figure legend and methods updated accordingly. The text regarding this update now reads:

Line 188-195: "To qualitatively compare the binding poses of D-ribulose versus L-arabinose, we used the crystal structure of AauA to perform molecular docking of L-arabinose into its binding pocket using AutoDock Vina. Six different conformations of L-arabinose were docked into the binding site and while all conformers occupied a similar binding position as D-ribulose, the predicted binding mode of L-arabinose exhibited fewer contacts within the substrate pocket of AauA."

REVIEWERS' COMMENTS

Reviewer #2 (Remarks to the Author):

My comments and concerns have been well addressed, particularly with the improved purity of the kinase. Please note, however, that there is a mistake in Figure S6: panels b and c have been reversed, and one trace is missing in panel d. Otherwise, this is a very nice paper. Congratulations on this impressive work.

Jean-Michel Jault

Thank you for positive feedback. Figure panels b and c have been correctly indicated in the legend and manuscript text. Please note there is no missing trace on panel d –

the glucose trace did not shift from that of the buffer only control and as such the two lines overlap. This has been stated in the legend.